# Human Serum Albumin-Based Nanoparticles for Targeted Intracellular Drug Delivery

**DOI:** 10.3390/ijms26178297

**Published:** 2025-08-27

**Authors:** Claudia Gabriela Chilom, Sorina Iftimie, Adriana Elena Balan, Daniela Oprea, Monica Enculescu, Teodor Adrian Enache

**Affiliations:** 1Faculty of Physics, University of Bucharest Magurele, 077125 Magurele, Ilfov, Romania; claudia.chilom@fizica.unibuc.ro (C.G.C.); sorina.iftimie@fizica.unibuc.ro (S.I.); adriana.balan@unibuc.ro (A.E.B.); daniela.oprea@infim.ro (D.O.); 2National Institute of Materials Physics, Str. Atomistilor, nr. 405A, 077125 Magurele, Ilfov, Romania; mdatcu@infim.ro

**Keywords:** human serum albumin nanoparticles, folic acid targeting, rutin delivery system

## Abstract

We report the synthesis and characterization of folic acid (FA)-conjugated human serum albumin nanoparticles, (HSA-FA):Ru NPs, as targeted carriers for rutin (Ru), a flavonoid with known anticancer activity. Nanoparticles were fabricated via a desolvation method, and their surface was functionalized with folic acid to promote selective uptake by cancer cells overexpressing folate receptors. Morphological and dimensional analyses performed by atomic force microscopy (AFM), scanning electron microscopy (SEM), and fluorescence microscopy confirmed that all nanoparticles were below 100 nm and exhibited good colloidal stability. Voltametric measurements confirmed the successful incorporation of both rutin and folic acid within the (HSA-FA):Ru nanoparticle formulation. Biological evaluation was conducted on healthy L929 fibroblasts and HT-29 colon adenocarcinoma cells. MTS colorimetric assays revealed that (HSA-FA):Ru NPs significantly reduced the viability of HT-29 cells, while maintaining higher compatibility with L929 cells. Fluorescence and electron microscopy further confirmed preferential nanoparticle uptake and surface accumulation in HT-29 cells, supporting the role of folic acid in enhancing targeted delivery. The study demonstrates that HSA-based nanoparticles functionalized with FA and loaded with Ru offer a biocompatible and efficient strategy for selective intracellular drug delivery in colorectal cancer. These findings support the use of albumin-based nanocarriers in the development of targeted therapeutic platforms for cancer treatment.

## 1. Introduction

Colorectal cancer (CRC) is one of the leading causes of death worldwide regardless of gender or age. CRC occurs in patients under the age of 50 years and is associated with dietary habits that cause major imbalances in the gastrointestinal flora [1]. Five stages of CRC are known, from stage 0 (characterized by the presence of abnormal colon cells or polyps in the mucosa; it is completely curable by surgical resection) [2] to stage 4 (characterized by a survival rate of less than 6% in 5 years) [3]. The therapeutic efficacy of currently available treatments (chemotherapy, radiotherapy) is limited and is associated with adverse effects. Efforts are being made to discover new therapeutic agents, such as targeted drug delivery systems for CRC therapy, through which anticancer drugs can be delivered more efficiently.

Targeted delivery systems for drugs and other active ingredients largely depend on parameters such as drug solubility, stability, and permeability [4]. To improve this, nanoparticle-based therapeutics have been developed based on the ability of NPs to control the release of the encapsulating agent, target it, and reduce the number of doses administered to patients.

One way to tackle CRC is through the use of NPs. Their small size, continuous or controlled drug release, targeting of the drug to the desired cells/tissues, and the fact that they can be administered intravenously and circulate over a long period of time with reduced clearance rates from the renal system [5] are advantages that can be applied in CRC treatment protocols.

Many attempts have been made to create structures based on protein molecules such as serum proteins [6,7,8]. These structures have excellent biocompatibility and are biodegradable; therefore, they can be used for targeted delivery of various drugs or molecules of interest to a wide spectrum of cancer cells. In the last decade, several human serum albumin (HSA) drug delivery systems have been proposed for colon cancer treatment. One of these studies proposed HSA nanoparticles (HSA NPs) loaded with doxorubicin (DOX) and an antitumoral protein (TRIAL) (NPs size being about 120 nm) as vehicles for the delivery of DOX into colon tumor cells [9]. These NPs exhibited excellent antitumor efficacy in HCT-116 colon cancer due to the synergistic effect of the two tumor agents, TRIAL and DOX. HCT 116 colon cancer cells were also targeted by piperlongumine-loaded HSA NPs (PL NPs) [10]. These NPs showed sustained and prolonged drug release profile and increased the anti-proliferative activity of piperlongumine against HCT 116 colon cancer cells. PEG-coated HSA NPs loaded with the cytostatic 5-Fluorouracil were obtained and tested on Hep29 human colon cell lines [11].

One modality to facilitate entry of serum protein nanostructures into cells [12] that encapsulate or are covered with drugs or other structures of medical interest is the use of folic acid (vitamin B9) (FA). Unlike healthy cells, cancer cells have cellular receptors for FA, such as the folate receptor (FR) [13], a validated biomarker for tumor cells because of its overexpression in various tumors. FA appears to play an important role in the etiology of digestive tract cancers, including colorectal, esophageal, and stomach tumors [14,15] and has a chemopreventive effect in colorectal cancer [16]. A diet deficient in FA has been associated with an increased risk of colon cancer, and supplementation with FA provides significant protection against the disease [15]. Some studies suggest that FA inhibits the growth of colon cancer by proliferating anti-cancer and anti-angiogenesis cells [17]. The choice of FA as a component part of synthesized NPS is based on its effects against colon cancer, as well as its affinity for specific cellular receptors (FR), [13] in the membranes of cancer cells.

Some flavonoids inhibit the proliferation of CRC cells [18] or suppress CRC metastasis [19] by modulating several signaling pathways. There is evidence that flavonoid rutin (Ru) has anticancer properties against human colon cancer (HCT 116) cell lines using an in vitro method [20]. Current technological approaches can pave the way for the design and use of new, more sophisticated drug conjugate structures, especially since FR is targeted through the use of several complementary technologies: small molecules, NPs, and proteins—thus providing broad and distinct knowledge in the field. Among these, rutin (Ru) is a flavonoid that has previously been characterized in interaction with albumin molecules [21,22], but also in nanohybrid formulations [8].

The choice of method for preparing nanoparticles for drug delivery purposes should take into account the type of NPs and the ability to control the experimental parameters. Protein-based NPs have advantages such as biodegradability, biocompatibility, low toxicity, but also properties such as high absorption, colloidal stability, self-assembly, and flexibility in surface modification, allowing the attachment of ligands of interest—drugs, contrast agents, or targeting ligands [23]. Among the methods described in the literature for the preparation of protein NPs (self-assembly, emulsification, thermal gelation, nanospray-drying, and desolvation) [24], the desolvation method allows the control of the experimental conditions and the adaptation of the parameters, such as pH, temperature, ionic straight, nature and amount of dissolving agent, or concentration of cross-linking agent and drug content, to obtain stable NPs of the desired shape and size for biomedical applications.

In this study, Ru was chosen to target L929 fibroblast cell lines and HT-29 adenocarcinoma human as a potential replacement for chemical therapeutic drugs. We developed HSA-based biohybrids (HSA-FA):Ru NPs for targeted delivery of Ru into HT-29 and L929 cells. Tumor cell targeting is achieved via folic acid (FA) intermixing, for which tumor cells have membrane receptors. The development of (HSA-FA):Ru nanohybrid is part of the efforts to improve CRC therapies and minimize severe adverse effects associated with cytotoxic drugs.

## 2. Results and Discussion

The focus of this work was the development and characterization of a folic acid-functionalized, HSA-based drug delivery system and its preliminary in vitro evaluation. Once the albumin nanoparticles loaded with rutin are internalized by cells, they are exposed to cytosolic proteases, which facilitate the release of the Ru load [25,26].

### 2.1. Characterization of HSA and (HSA-FA):Ru Nanoparticles

Several methods were used to characterize the physicochemical properties of the HSA and (HSA-FA):Ru nanoparticles: AFM and SEM for morphological characterization, fluorescence microscopy to evaluate aggregation into amyloid structures, and cyclic voltammetry to evaluate the Ru and folic acid load of NPs.

The AFM characterization of HSA and (HSA-FA):Ru nanoparticles is presented in Figure 1, including 3D topography images, corresponding phase contrast images over 5 µm × 5 µm scan areas, and histograms of maximum height distributions.

The topography reveals significant morphological differences in terms of surface homogeneity and particle sizes. Considering that the lateral dimension measured by AFM would be larger than the actual size because of the tip profile, maximum height measurements were considered for evaluating particle dimensions. The HSA globular structures have a narrow size distribution, with a maximum height mean value of 14.25 ± 1.40 nm, while (HSA-FA):Ru particles range in the interval 21.70 ± 7.76 nm. According to literature, HSA NPs dimensions are highly dependent on the processing conditions, such as pH, concentration, temperature or desolvation agent [27]. It is expected that small-sized nanoparticles enhance tissue penetration, making HSA and (HSA-FA):Ru NPs suitable for targeted delivery of Ru into cancer cells [28,29].

HSA NPs and (HSA-FA):Ru NPs were also analyzed by SEM. The two types of NPs were deposited on conductive surfaces (1 cm^2^) of Au-coated glass. The use of conductive surfaces was preferred to any other type of surface, due to the advantage of decreasing the degree of polarization as a result of the accumulation of charges on the surface during electron beam usage. For the HSA NPs, the SEM images obtained (Figure 2A) showed a uniform distribution, with these NPs having small sizes of about 50–70 nm. Similarly, (HSA-FA):Ru (Figure 2B) showed sizes in the tens of nanometers range, while forming aggregates in the hundreds of micrometers range.

The sizes obtained by SEM for the two types of NPs were slightly larger than those measured by AFM. This discrepancy may be attributed to: (1) the different types of surfaces on which the NPs were deposited, (2) the distinct nature of the parameters measured by each technique, or (3) sampling-related effects. Nevertheless, all measured sizes were below 100 nm, which supports their intended application for cellular interaction. Although the values obtained by SEM and AFM are not directly comparable, the two techniques provide complementary information regarding particle morphology and structure. Specifically, AFM measures particle height, while SEM determines the projected diameter.

Albumin possesses a secondary structure characterized by 10% content of β-sheet motifs, which are often retained during NPs formation. However, many studies have reported that albumins can form large aggregates resembling amyloid-like structures in vitro. A critical step in amyloid fibril formation is the destabilization of native conformation, typically induced by changes in pH, temperature, ionic strength, and other environmental factors [30,31]. Since the protocol used in this study avoided such destabilizing conditions—and in light of the stability data presented in the Appendix A—we investigated the effect of concentration on fibril formation using Thioflavin T as a fluorescence probe for fibrillization.

Thioflavin T (ThT) is a benzothiazole dye that exhibits minimal fluorescence in solution but undergoes a dramatic increase in fluorescence intensity upon binding to β-sheet-enriched structures, particularly those forming cross-β motifs commonly found in amyloid fibrils and similar aggregates [32]. The mechanism of ThT binding involves its insertion into the grooves formed by the repetitive β-sheet architecture, where restricted intramolecular rotation leads to enhanced quantum yield and fluorescence emission (excitation: ~440–450 nm; emission: ~480–490 nm). This fluorescence enhancement is widely used as an indicator of the presence and spatial distribution of supramolecular β-sheet structures. In this context, ThT labeling serves as a sensitive tool for confirming the structural organization of albumin nanoparticles and for their selective detection under fluorescence microscopy. Fluorescent labeling of albumin nanoparticles with Thioflavin T enables visualization of β-sheet-rich domains by fluorescence microscopy. Thus, HSA-NPs and (HSA-FA):Ru NPs were labeled with the fluorescent indicator thioflavin T (ThT), and labeling was performed by incubating NPs of concentration 200 mg/mL with 1 mM thioflavin T in a working volume of 1 mL, followed by centrifugation and washing. For fluorescence assessment, HSA NPs and (HSA-FA):Ru NPs were deposited on a glass surface. From fluorescence microscopy images (Figure 3), it was observed that both HSA NPs (Figure 3A) and (HSA-FA):Ru NPs (Figure 3B) formed large aggregates and tended to structure into amyloid-like structures.

Similar results obtained by laser diffraction analysis, on both samples, revealed the presence of a high-abundance population with particle sizes around 6–7 µm in both nanoparticle systems, Appendix A. Importantly, precursor aggregates, corresponding to the initial nanoparticles, were still detected, with a size distribution below 100 nm. This observation suggests that, even after prolonged incubation, a fraction of the nanoparticles retains their original dimensions, while a significant proportion undergoes aggregation into larger structures, with differences that (HSA-FA):Ru presented aggregates around 10 µm compared with the sample without Ru, where the magnitude rich less than 500 µm, Appendix A.

Considering the observed aggregation phenomena, special attention was given to stock solution of NPs considering the differences in NPs concentration used in the uptake studies. While aggregation studies—which likely involve alterations in tertiary structure, nucleation, and fibril growth—were conducted at significantly higher concentrations, the cell lines were incubated with nanoparticle concentrations up to 2 mg/mL.

Rutin is a flavonoid glycoside (quercetin-3-rutinoside) that contains two distinct electroactive moieties: a catechol group, which undergoes a reversible oxidation around +0.25 V (vs. Ag/AgCl), and a resorcinol group, which is irreversibly oxidized at approximately +0.65 V (vs. Ag/AgCl) [31]. The redox behavior of folic acid is primarily determined by its pteridinic ring system, which is involved in irreversible reduction processes at highly negative potentials through a two-electron, two-proton transfer mechanism [33]. Additionally, folic acid can also be oxidized at more positive potentials, typically above +0.9 V (vs. Ag/AgCl) [34]. Based on these redox characteristics, both rutin and folic acid are suitable for detection and quantification using voltametric techniques.

It should be noted that albumin NPs possess redox-active amino acid residues such as cysteine, tyrosine, tryptophan, histidine, and methionine. At physiological pH, the redox charge transfer associated with these side chains occurs at oxidation potentials typically above +0.5 V vs. Ag/AgCl [35,36]. Thus, to avoid interference from the redox-active amino acid residues, the electrochemical evaluation of Ru and folic acid within the NPs system was conducted by cyclic voltammetry within selected potential windows: 0.0 to +0.4 V (vs. Ag/AgCl) for rutin, and −0.2 to −1.0 V (vs. Ag/AgCl) for folic acid. Measurements were performed at pH 7.0 in 0.1 M phosphate buffer, using a 4 mg·mL^−1^ dispersion of (HSA-FA):Ru NPs (200 mg mL^−1^ stock solution; dilution factor 1:50) (Figure 4). Taking into account that only molecules in direct contact with the working electrode contribute to the redox response—making absolute quantification challenging in the NPs system—relative concentrations of both Ru and folic acid were determined using the double standard addition method.

On the first going scan, obtained in solution of 4 mg mL^−1^ (HSA-FA):Ru NPs, between the 0.0 V to 0.4 V one oxidation peak, at *E_pa_* = +0.26 V occurred, (Figure 4A1). Reversing the scan direction, a new cathodic peak appeared around the *E_pc_* = 0.22 V (Figure 4A1), and the difference between the anodic and the cathodic peaks was close to the theoretical value of 30 mV for a two-electron reversible reaction. Moreover, the addition of rutin standards resulted in redox peaks at similar potentials, with progressively higher current responses. These observations strongly indicate that the detected redox couple corresponds to the electroactive catechol moiety of Ru. Based on the standard addition plot (Figure 4A2), a Ru concentration of 2.86 µM (1.75 µg·mL^−1^) was determined in the electrochemical cell. Considering the 1:50 dilution factor, this corresponds to a concentration of 143 µM (87.5 µg·mL^−1^) in the synthetized stock solution. This value reflects the fraction of rutin molecules exposed on the surface of the Nps and which are available for electrochemical oxidation. Given the substantial difference between the obtained concentration and the 3.33 mg·mL^−1^ Ru used during NPs synthesis, it can be inferred that Ru is predominantly encapsulated within the albumin NPs [37], rather than surface-bound. Although the rutinoside sugar moiety of Ru is polar, the hydrophobic character of the aromatic rings in its aglycone part, quercetin, facilitates its incorporation into hydrophobic matrices such as albumin NPs.

Similar electrochemical analyses were conducted to evaluate FA content within the NPs system. On the first anodic going scan, from −0.2 V till −1.0 V, recorded in 4 mg mL^−1^ (HSA-FA):Ru NPs, at pH = 7.0 and 0.1 M phosphate buffer, at glassy carbon electrode, a single reduction peak was observed at approximately −0.9 V, corresponding to the electrochemical reduction of FA. Returning to more positive values, from −1.0 V to −0.2 V, an oxidation peak, corresponding to the oxidation of the reduced products of FA, was observed, (Figure 4B1). Taking the reduction peak of folic acid as the analytical signal for quantitative evaluation and performing cyclic voltammetry with two successive standard additions in the same solution, a concentration of 36.45 µM (16.1 µg mL^−1^)—corresponding to the fraction of FA bound to the albumin NPs—was determined in the electrochemical cell. Considering the 1:50 dilution factor of the (HSA-FA):Ru NPs and the initial concentration of 6.66 mg·mL^−1^ folic acid used in the synthesis, the calculated concentration of 1.82 mM (4.37 mg·mL^−1^) corresponds to 65.61% of the precursor amount in the synthesized stock solution. Unlike Ru, which is predominantly encapsulated within the albumin NPs, FA is bounded on the NP surfaces, being accessible for oxidation and accurate quantification.

### 2.2. Assessment of Cell Viability of HT-29 After Exposure to Nanoparticles

The HSA-FA: Ru NPs system was designed to be used as a tumor-targeting vector, including carrying a therapeutic agent. In this context, Ru serves as the therapeutic cargo. The covalent conjugation of NPs with folic acid enables targeted delivery by binding to folate receptors of tumor cell membranes. As the cancer cells express much more folic acid receptors than normal cells, folic acid-conjugated NPs can be used for targeted drug delivery, enhancing treatment efficacy while minimizing side effects on healthy tissues.

In order to evaluate the colon cancer targeting effect of HSA NPs and (HSA-FA):Ru NPs, cellular viability and imaging studies were assessed for a concentration range from 0 till 2 mg mL^−1^. The maximum concentration tested in vitro (2 mg/mL) allows the full spectrum of biological response to be characterized, including effects at high doses. In vivo, the effective concentration at tumor level is significantly influenced by biodistribution, volume of administration, and systemic clearance. Although the concentration of 2 mg/mL does not directly reflect a physiological dose, it provides useful information for assessing therapeutic potential. For in vivo applications, pharmacokinetic and biodistribution studies will be required.

Cell viability studies (Figure 4) were performed using the MTS protocol and were complemented by fluorescence microscopy (Figure 5) and SEM (Figure 6). The results obtained for the HT-29 human colon adenocarcinoma cell line following the application of the two types of NPs, HSA NPs and (HSA-FA):Ru NPs, were compared with those obtained for a healthy cell line, the L929 fibroblast line.

The viability of the two cell lines, L929 and HT-29, subjected to the action of HSA NPs, was good, with values above 90%, compared to the control (cells not incubated with HSA NPs), as shown in Figure 5A (■) and 5B (■). Although HSA shows properties such as biodegradability, biocompatibility, and low toxicity, our observations suggest that under certain conditions, HSA NPs may have notable biological effects that must be clarified in future investigations. By using (HSA-FA):Ru NPs, a slight decrease of viability, but still elevated, was obtained for the L929 line, about 70.66% (Figure 5A (●), for NPs concentrations above 1 mg/mL, whereas higher concentrations (around 2 mg/mL) of NPs induced a decrease of viability, at 54.58% (Figure 5B (●), for the HT-29 cell line. These results can be explained by the presence of FA in the rutin-loaded NPs structure, which allows an intracellular delivery of active compounds, given that the surface of cancer cells expresses, as mentioned above, a high number of folic acid receptors.

The effect of cellular uptake of both types of NPs was also investigated using L929 and HT-29 cells lines by the means of SEM and fluorescence microscopy, as shown in Figure 6 and Figure 7. SEM images of L929 cells recorded before different experimental conditions (Figure 6A1–C1) demonstrate that exposure to the NPs did not induce any noticeable morphological changes in fibroblasts. In contrast, SEM images of HT-29 cell clusters (Figure 6A2–C2) revealed a significant reduction in cell size following incubation with (HSA-FA):Ru NPs (Figure 6C2) compared to the untreated control group (Figure 6A2), suggesting a possible cytotoxic or stress-related response specifically in the targeted cancer cells. It should be noted that the HT-29 cells are much smaller than fibroblasts and tend to grow in clusters.

For fluorescence microscopy evidence of the effect of HSA NPs and (HSA-FA):Ru NPs on the two cell types, L929 and HT-29, NPs were labeled with thioflavin T (green). ThT is a marker whose fluorescence is used as an indication of the presence of amyloid structures [38]. ThT exhibits a strong fluorescence upon binding to amyloids [39]. HSA NPs are observed to be evenly distributed on the cell surface, but also as adsorbed structures on the surface. The NPs pathway in the cell can be visualized as green spots [40]. Therefore, we can say that the NPs were directed towards cytoskeletal proteins with high β-structure content.

DAPI is a fluorescent dye that labels DNA and allows easy visualization of the nucleus in interphase cells and chromosomes in mitotic cells [40]. DAPI specifically stains double-stranded DNA without non-specific labelling in the cytoplasm. After binding to DNA, DAPI fluorescence increases 20-fold. The fluorescent dye DAPI (blue) was used to visualize the cell nucleus (Figure 6).

Similar to viability results (Figure 5), for HSA NPs only a small decrease on cells number was observed in both lines (Figure 7B1,B2) when compared with control experiments (Figure 7A1,A2), whereas the cell morphology remains the same. However, a selective distribution was observed for (HSA-FA):Ru NPs, with regions of high cell density being preferred. It is possible that such closeness leads to the expression of folic acid receptors more obvious in the case of HT-29, were the tendency for NPs to clump on the cell surface was observed (Figure 6C1,C2), with a more pronounced effect for (HSA-FA):Ru NPs (Figure 7C2). In the case of L929 cells both the nucleus and cytoskeleton were exposed to (HSA-FA):Ru NPs and exhibit similar morphology to the control. Also, both lines presented a larger nucleus than control cells. In addition, for HT-29, a decrease of clusters number and size was observed after incubation with both NP systems, with a highest decrease induced by (HSA-FA):Ru NPs.

Therefore, HSA NPs target both at the level of the nucleus, by acting on DNA, and indirectly, by targeting the cytoskeletal proteins with high β-structure content. Direct connections between the actin cytoskeleton and the nucleus may be damaged, which will have implications on nuclear positioning, gene expression, or damage to the nuclear envelope [41].

The focus of this work was the successful development and characterization of a folic acid-functionalized HSA-based delivery system and its preliminary evaluation in vitro. Future studies will address the optimization of NPs formulations and treatment conditions, aiming to define therapeutic concentration windows that ensure maximum cytotoxicity toward cancer cells while preserving normal cell viability. Our current results provide strong evidence of the differential cytotoxic effect of (HSA-FA):Ru NPs-marked by a significant reduction in HT-29 cell viability and minimal impact on L929 fibroblasts.

## 3. Materials and Methods

### 3.1. Materials

Human serum albumin (HSA, ≥98%), glucose (180.15 Da), thioflavin T (318.86 Da), and ethanol (EtOH, 46.07 Da, ≥99.8%) were purchased from Merck Company (Darmstadt, Germany). Folic acid (FA, 441.40 Da, ≥97%) was purchased from Fluka AG (Buchs, Switzerland). Rutin (Ru, 610.52 Da, ≥97%) was purchased from ACROS ORGANICS (Geel, Belgium), 1-ethyl-3-(dimethylaminopropyl) carbodiimide (EDC, 191.7 Da), 4′,6-diamidino-2-phenylindole (DAPI) was purchased from Thermo Fisher Scientific, Waltham, MA, USA, and N-hydroxysuccinimide (NHS, 115.09 Da, ≥98%)) was purchased from Thermo Fisher Scientific (Waltham, MA, USA). Formaldehyde (36.5–38%), glutaraldehyde (Grade I, 25%), and osmium tetroxide (OsO_4_ 3.5–4.5%) were acquired from Sigma-Aldrich, Merck, Germany. Dulbecco’s Modified Eagle Medium (DMEM, Thermo Fisher Scientific, Waltham, MA, USA), fetal bovine serum (FBS, Thermo Fisher Scientific, Waltham, MA, USA), phosphate buffer saline without calcium and magnesium (PBS, Thermo Fisher Scientific, Waltham, MA, USA), 0.25% trypsin/EDTA solution (Thermo Fisher Scientific, Waltham, MA, USA), and penicillin–streptomycin (Thermo Fisher Scientific, Waltham, MA, USA). The MTS (3-(4,5-dimethylthiazol-2-yl)-5-(3-carboxymethoxyphenyl)-2-(4-sulfophenyl)-2H-tetrazolium) kit was also purchased from Thermo Fisher Scientific, Waltham, MA, USA. Phalloidin-iFluor™ 647 Conjugate, Phalloidin-iFluor™ 488 Conjugate from Cayman Chemical, FluorSave™ Reagen from Merck Millipore. Two cell lines were utilized, L929 F10 fibroblasts cells and HT-29 colorectal adenocarcinoma cells, both cell lines were purchased from ATCC, (Manassas, VA, USA).

### 3.2. Preparation of HSA NPs

In the first step, HSA-NPs were synthesized by the desolvation method previously described for the preparation of bovine serum albumin nanoparticles [8]. HSA NPs were obtained at room temperature, using EtOH as desolvation agent and glucose as cross-linking agent. Briefly, the pH of 12 mL HSA (200 mg mL^−1^) solution was established at 8.15, then the sample was stirred (CRS 15X CAPP, Odense, Denmark) at 550 rpm for 10 min. After that, 8 mL EtOH were dropwise (1 mL min^−1^), until the suspension containing 120 mg mL^−1^ HSA became cloudy. A volume of 350 µL glucose (corresponding to 8% final concentration) was added to this suspension, and it was left for 12 h under continuous stirring at 550 rpm. This sample was centrifuged three times (10,000× *g*, for 10 min), each centrifugation being followed by ultrasonication in an ultrasound bath (BRANSON 1210, Marshall Scientific, Hampton, NH, USA) for 5 min. After each centrifugation step, the pellet was resuspended in 100 mL of distilled water (24 mg mL^−1^ HSA-NP). When necessary, the NP solutions were concentrated by centrifugation.

### 3.3. Preparation of (HSA-FA)

Ru loaded nanoparticles. HSA (12 mL of 200 mg mL^−1^ at pH 8.15) solution was stirred at 550 rpm for 10 min. Rutin (Ru, 40 mg mL^−1^) was dissolved in 8 mL EtOH and was added dropwise (1 mL min^−1^), until the suspension containing 120 mg mL^−1^ HSA became cloudy. A volume of 350 µL glucose (corresponding to 8% final concentration) was added to this suspension, and it was left for 12 h under continuous stirring at 550 rpm. Meanwhile, 0.5 mL of 160 mg mL^−1^ FA was activated with EDC (0.554 mM) and NHS (0.554 mM) for 2 h under continuous stirring at 550 rpm and in the dark. Activated FA was added over HSA-Ru suspension, at a concentration of 16 mg mL^−1^, under continuous stirring for 2 h. The resulting (HSA-FA):Ru nanoparticle dispersion was purified by three consecutive centrifugation steps (10,000× *g*, 10 min each), with each step followed by 5 min of ultrasonication. After each centrifugation, the pellet was resuspended in 100 mL of distilled water (24 mg mL^−1^ (HSA-FA):Ru NPs). For aggregation and voltametric studies, the stock solutions were kept at 200 mg mL^−1^.

### 3.4. Cell Cultivation

The cellular proliferation process was conducted within a controlled laboratory setting using DMEM culture medium enriched with 10% fetal bovine serum, 4.5 g/L glucose, 2 mM L-glutamine, penicillin (100 U/mL), and streptomycin (100 μg/mL). The cells were incubated in a specific environment at a temperature of 37 °C, 5% CO_2_ atmosphere, and high humidity. Subsequent cell passaging was performed using T-25 cell culture flasks until reaching an approximate cell confluence of 80%. For fluorescence analysis, cellular fixation was achieved by immersing the cells in a 4% formaldehyde solution for 15 min at room temperature. Additionally, for scanning electron microscopy (SEM) analysis, cell fixation was accomplished using a solution containing 4% formaldehyde and 0.25% glutaraldehyde. Subsequently, an additional treatment involving 0.1% osmium tetroxide incubation was applied to the fixed cells for 20 min at room temperature for SEM imaging purposes.

The L929 cell line was subjected to staining with DAPI for nuclear visualization and Alexa Phalloidin-iFluor™ 488 Conjugate to visualize the cytoskeletal structure in fluorescence analysis. On the other hand, HT-29 cells were stained with Phalloidin-iFluor™ 647 Conjugate to visualize the cytoskeleton and DAPI for nuclear staining. Post-staining, the specimens were mounted on microscopic slides using FluorSave™ Reagent, from Merck Company (Darmstadt, Germany), and observed using a fluorescence microscope manufactured by Leica Microsystems.

### 3.5. Scanning Electron Microscopy (SEM)

The morphology of the obtained particles samples was investigated with a Zeiss Gemini SEM 500 field-emission scanning electron microscope (FESEM), working in both High Vacuum (HV) and Variable Pressure (VP) modes, from 0.2 to 30 kV, equipped with InLens and SE2 detectors. Samples were prepared on silica substrates by free adsorption. A volume of 20 µL of a 1 mg mL^−1^ solution was deposited onto the surface and allowed to adsorb for 5 min, after which the substrates were gently washed to remove unbound material.

### 3.6. Atomic Force Microscopy (AFM)

Three-dimensional topography and phase contrast images were obtained using SPM-NTegra Prima AFM equipment (NT-MDT, Moscow, Russia) operated in semi-contact mode, as recommended for soft samples. Measurements were performed in air at room temperature, using a NSG 01 cantilever with a resonant frequency between 87 kHz and 230 kHz and a constant force between 1.45 N/m and 15.1 N/m. All samples were deposited on freshly cleaved mica and dried at room temperature. Images were processed and analyzed by means of NT-MDT Image Analysis 2 software version 2.1.2. Samples were prepared on mica substrates by free adsorption. A volume of 20 µL of a 1 mg mL^−1^ solution was deposited onto the surface and allowed to adsorb for 5 min, after which the substrates were gently washed to remove unbound material.

### 3.7. Fluorescence Microscopy

The fluorescence images were obtained with Leica DM6B upright fluorescence microscope (Leica Microsystems CMS GmbH, Wetzlar, Germany) equipped with a Leica CTR6 LED and a Leica EL6000 external light source. The 40X objective (0.65 NA, 0.36 mm WD and correction ring) from Leica was used for sample analyzation with excitation filter cube 480/50 nm, dichroic mirror 505–510 nm, and emission filter 527/30 nm equipped with 4.2 MP sCMOS Leica DFC9000 monochrome fluorescence camera.

### 3.8. Voltammetric Measurements and the Electrochemical Cell

The voltammograms were achieved using an IVIUM potentiostat in combination with IviumSoft program version 2.219 (Ivium Technologies, Eindhoven, The Netherlands). All the measurements were carried out in a one-compartment 2 mL electrochemical cell, using a working glassy carbon electrode (GCE) (d = 1.6 mm), a counter Pt wire, and an Ag/AgCl (3 M KCl) reference electrodes. All cyclic voltammograms were recorded at a scan rate of 50 mV s^−1^ with a step potential of 2 mV. Before each experiment the GCE was polished using diamond spray (particle size 1 µm) on a microcloth pad. After polishing, the electrode was rinsed thoroughly with Milli-Q water for 30 s; then it was placed and pretreated in the supporting electrolyte where various cyclic voltammograms were recorded until a steady state baseline voltammogram was obtained. Prior to recording any voltammograms in the negative potential range, the solutions were purged with nitrogen to remove dissolved oxygen. Control voltammograms recorded without deoxygenation exhibited a distinct reduction peak at −0.6 V (vs. Ag/AgCl).

Due to their redox-active properties, both rutin and folic acid are suitable for detection and quantification using voltametric techniques. The standard addition method was employed to determine the unknown concentration of the analyte, as shown in Figure 4 ((A) rutin and (B) folic acid). Therefore, the concentrations of red curves have to be determined, and this was carried out by plotting the currents for both standards (A2 and B2). The OX intercept for each plot represents the unknown concentration.

### 3.9. Cell Viability

The cell viability was investigated using the MTS assay based on conversion of the tetrazolium salt MTS (3-(4,5-dimethylthiazol-2-yl)-5-(3-carboxymethoxyphenyl)-2-(4-sulfophenyl)-2H-tetrazolium) to a purple formazan in the presence of phenazine methosulfate. The MTS assay procedure involved adding 10 μL MTS reagent to each incubated sample, and after 4 h, the culture medium was collected and the absorption was measured at 490 nm using a plate reader FLUOstar Omega, BMG Labtech, Ortenberg, Germany.

### 3.10. Statistical Analysis

Data are presented as the mean  ±  standard deviation (SD). The statistical significance of differences between experimental groups was calculated using One-way analysis of variance with Tukey’s Multiple Comparison test. The values of *p*  <  0.05 were considered statistically significant.

## 4. Conclusions

In summary, a folic acid-conjugated human serum albumin nanoparticles (HSA-FA):Ru NPs drug delivery system was synthesized to deliver Ru to L929 fibroblast cell lines and HT-29 human adenocarcinoma. After functionalization with FA, cyclic voltammetry was used to evaluate the incorporation of both rutin and folic acid within the (HSA-FA):Ru nanoparticle formulation, and the morphological properties of NPs were analyzed by AFM and SEM. The dimensions of the NPs determined were smaller than 100 nm. Therefore, these NPs represent a valuable approach for the penetration of the membrane and for cellular target. This result, together with the time stability of the NPs, provided the basis for in vitro studies monitoring the effect of these NPs on L929 and Ht-29 cells.

The cytotoxicity of the HSA NPs and (HSA-FA):Ru NPs was evaluated by MTS assay using L929 and HT-29 cells after 24 h of incubation with the NPs. Following treatment of both cell types with NPs, the viability of HT-29 cells was more affected than that of L929 cells, and the effect was more pronounced in the presence of (HSA-FA):Ru NPs.

Fluorescence microscopy showed that HSA NPs and (HSA-FA):Ru NPs do not affect the nucleus and cytoskeleton of L929 and HT-29 cells, but for HT-29 cells a tendency of clusters shrinking was observed, and the effect was more evident for (HSA-FA):Ru NPs. while cyclic voltammetry was used to evaluate the incorporation of both Ru and FA within the (HSA-FA):Ru NPs formulation.

Taken together, the results of this study about the properties and the effect of HSA NPs and (HSA-FA):Ru NPs L929 and HT-29 cells could serve as guidelines in the rational design of proteins NP for drug delivery to cancer cells.

## Figures and Tables

**Figure 1 ijms-26-08297-f001:**
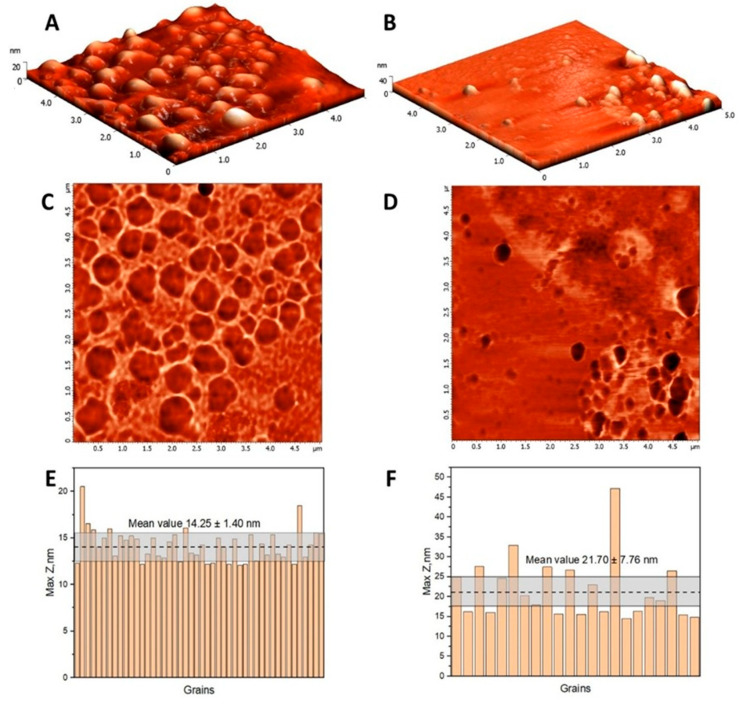
The topography images of HSA (**A**) and (HSA-FA):Ru-obtained particles (**B**) in semi-contact mode on a 5 µm × 5 µm surface areas, and the corresponding phase contrast images (**C**,**D**). Maximum height histogram for HSA (**E**) and (HSA-FA):Ru-obtained particles (**F**) resulting from the grain analysis of the topography images, where the dotted line represents the mean value and the corresponding confidence interval.

**Figure 2 ijms-26-08297-f002:**
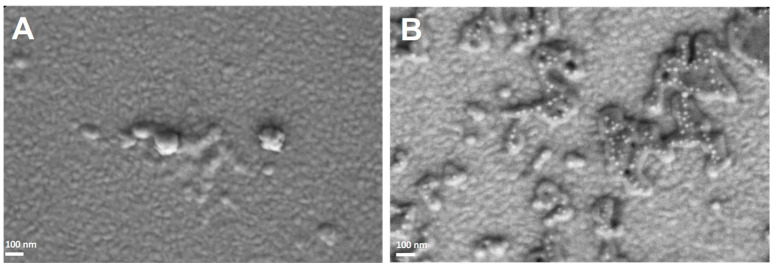
SEM images (100.00 kX) obtained for HSA-NPs (**A**) and (HSA-FA):Ru NPs (**B**) deposited on a conductive surface (Au-coated glass).

**Figure 3 ijms-26-08297-f003:**
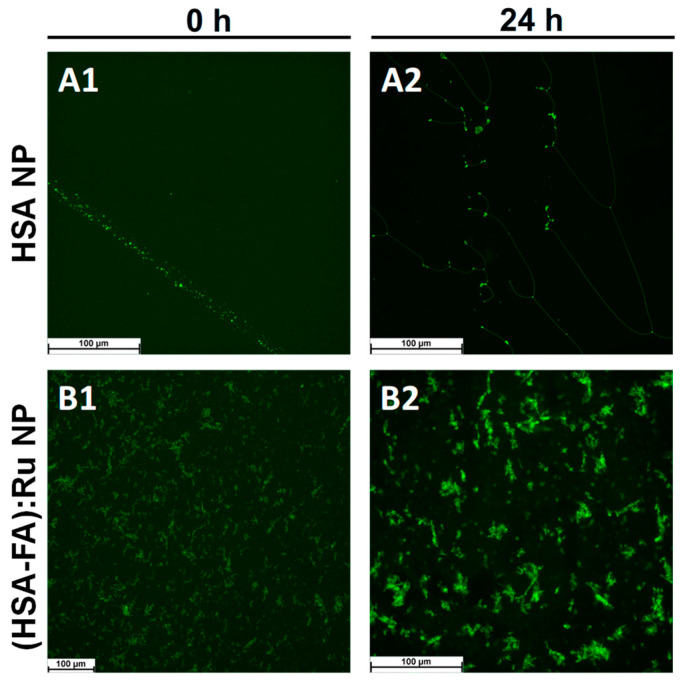
Fluorescence microscopy images obtained for (**A**) HSA NPs and (**B**) (HSA-FA):Ru NPs labeled with thioflavin T (ThT) after (**A1**) and (**B1**) at 0 h and (**A2**) and (**B2**) 24 h incubation, respectively, in aqueous solution.

**Figure 4 ijms-26-08297-f004:**
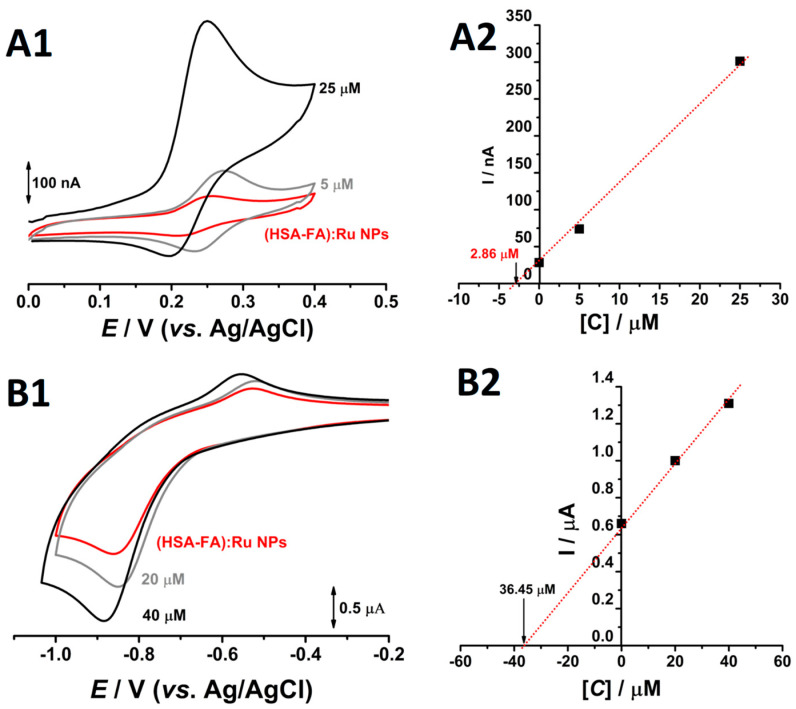
Cyclic voltammograms obtained at pH = 7.0 0.1M phosphate buffer, at glassy carbon electrode, for a solution of (**—**) 4 mg mL^−1^ (HSA-FA):Ru NPs in the presence of two different concentrations of (**A1**) rutin and (**B1**) FA and the corresponded plots of standard additions for (**A2**) Ru and (**B2**) FA.

**Figure 5 ijms-26-08297-f005:**
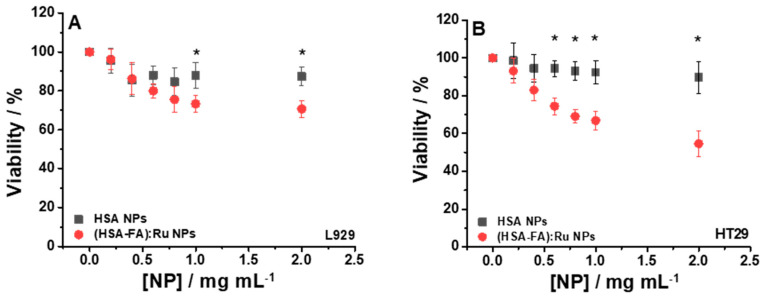
Cell viability relative to control for L929 (**A**) and HT-29 (**B**) cells incubated 24 h with different concentrations of HSA NPs and (HSA-FA):Ru NPs ((0.2–2) mg mL^−1^). For statistical analyses, ANOVA with Tukey post hoc tests were performed; *—*p* < 0.05.

**Figure 6 ijms-26-08297-f006:**
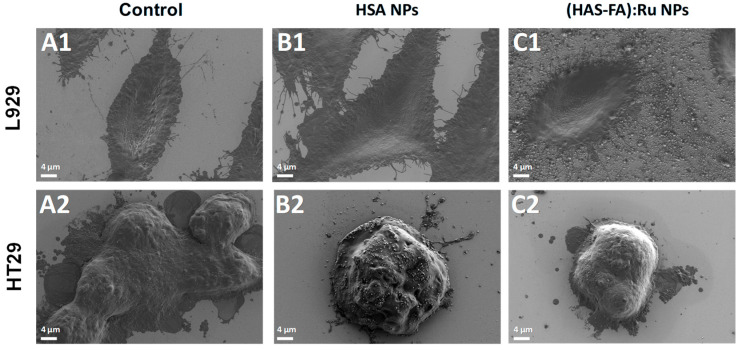
SEM images obtained for L929 and HT-29 cells recorded (**A1**,**A2**) before, and after 24 h incubation with (**B1**,**B2**) HSA NPs L929 and (**C1**,**C2**) (HSA-FA):Ru NPs.

**Figure 7 ijms-26-08297-f007:**
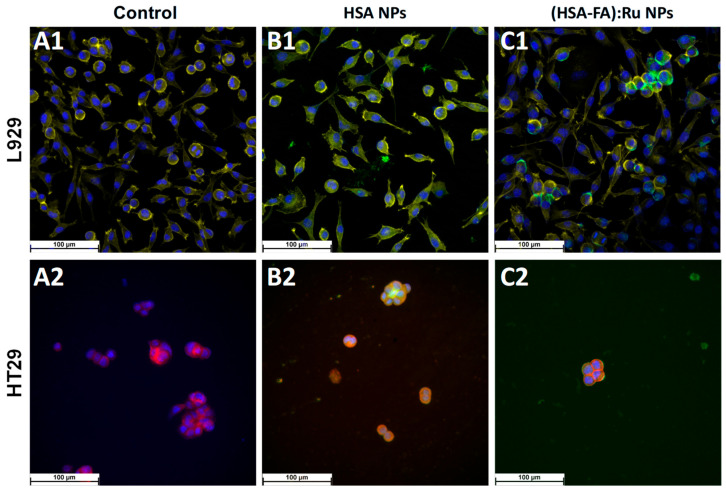
Fluorescence microscopy images obtained for L929 and HT-29 cells recorded (**A1**,**A2**) before and after 24 h incubation with (**B1**,**B2**) HSA NPs L929 and (**C1**,**C2**) (HSA-FA):Ru NPs.; (blue—DAPI; yellow (for L929), and red (for HT29)—phalloidin; green—ThT).

## Data Availability

The data presented in this study are available on request from the corresponding author.

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
