# Peer review of "Human Serum Albumin-Based Nanoparticles for Targeted Intracellular Drug Delivery"

_ijms, 2025, doi:10.3390/ijms26178297_

Round 1
Reviewer 1 Report (New Reviewer)
Comments and Suggestions for Authors
The higher fluorescence of ThT bound to HSA-FA:Ru NPs due to a high HSA b-sheet content should be confirmed by circular dichroism (CD). An increased b-sheet conformation of HSA might affect the cancer cellular uptake of the HSA-FA:Ru NPs reducing their uptake. It is therefore important to assess whether the b-sheet content can also be reduced. The size of the NPs claimed to be about 21nm, yet forming aggregates by fluorescence in the 100 of microns. Controlling the aggregations would certainly improve their cellular uptake.
Parallel to Voltammetric assay results, it is important to determine the content of FA linked to HSA and assess the reproducibility of HSA-FA:Ru NPs preparation by analysing different batches with UV and CD. The Abs and CD spectral features of FA and Ru do not overlap with those of HSA at wavelengths greater than 300nm (from the literature, both FA and Ru show Abs at about 350nm.
For these reasons I recommend a major revision for this manuscript.
Here is the list of questions to address.
Page 1, line 14 – I read “. functionalized with folic acid to promote selective uptake by cancer cells overexpressing folate receptors.” as the folate is covalently attached to the HSA.
Page 1, line 18-19 – If I understood correctly that folic acid is covalently link to HSA what about rutin? From “Voltametric measurements confirmed the successful incorporation of both rutin and folic acid within HSA-FA-Ru nanoparticle formulations” the name HSA-FA-Ru would indicate that both FA and Ru are covalently linked to HSA. However, as only FA is covalently linked to HSA as I assume it is, then the nanoparticle should be labelled (HSA-FA):Ru to avoid any confusing and misunderstanding.
Page 1, line 21 – Please add what MTS stands for and MTS to be included in the list of Abbreviations on page 13
Page 4, line 151 – The sentence “Albumin possesses a tertiary structure characterized by a high content of β-sheet motifs” is not correct. HSA contains a small amount of b-sheet, about 10%. If indeed your HSA-FA-Ru NPs contains high % of b -sheet, does this affect the cancer cellular uptake? Circular Dichroism of HSA-FA-Ru NPs would be the spectroscopic technique to assess the amount of b-sheet. If indeed your HSA-FA-Ru NPs does contain higher % of b -sheet, would an HSA-FA-Ru NPs with similar HAS secondary structure content be better? These questions need to be addressed.
Page 5, line 165 – labelling and not “abeling”
Page 5, line 168-170 - The sentence “From fluorescence microscopy images 168 (Figure 3) it was observed that both HSA NPs (Figure 3A) and HSA-FA-Ru NPs (Figure 169 3B) formed large aggregates and tended to structure into amyloid-like structures.” confirm that HSA-FA-Ru NPs appears to have much more ThT fluorescence consistent with higher b-sheet content than HSA-NP.
Page 6, line 199 - Not clear what are the concentrations of the red curves. Please add their values in the figures as well and clarify this in the legend.
Page 10, line 343 – Please include EDC and NHS in the list of Abbreviations and what do they stand for.
Page 11, line 401 - Please add the MTS supplier name.
Comments on the Quality of English LanguageThe higher fluorescence of ThT bound to HSA-FA:Ru NPs due to a high HSA b-sheet content should be confirmed by circular dichroism (CD). An increased b-sheet conformation of HSA might affect the cancer cellular uptake of the HSA-FA:Ru NPs reducing their uptake. It is therefore important to assess whether the b-sheet content can also be reduced. The size of the NPs claimed to be about 21nm, yet forming aggregates by fluorescence in the 100 of microns. Controlling the aggregations would certainly improve their cellular uptake.
Parallel to Voltammetric assay results, it is important to determine the content of FA linked to HSA and assess the reproducibility of HSA-FA:Ru NPs preparation by analysing different batches with UV and CD. The Abs and CD spectral features of FA and Ru do not overlap with those of HSA at wavelengths greater than 300nm (from the literature, both FA and Ru show Abs at about 350nm.
For these reasons I recommend a major revision for this manuscript.
Here is the list of questions to address.
Page 1, line 14 – I read “. functionalized with folic acid to promote selective uptake by cancer cells overexpressing folate receptors.” as the folate is covalently attached to the HSA.
Page 1, line 18-19 – If I understood correctly that folic acid is covalently link to HSA what about rutin? From “Voltametric measurements confirmed the successful incorporation of both rutin and folic acid within HSA-FA-Ru nanoparticle formulations” the name HSA-FA-Ru would indicate that both FA and Ru are covalently linked to HSA. However, as only FA is covalently linked to HSA as I assume it is, then the nanoparticle should be labelled (HSA-FA):Ru to avoid any confusing and misunderstanding.
Page 1, line 21 – Please add what MTS stands for and MTS to be included in the list of Abbreviations on page 13
Page 4, line 151 – The sentence “Albumin possesses a tertiary structure characterized by a high content of β-sheet motifs” is not correct. HSA contains a small amount of b-sheet, about 10%. If indeed your HSA-FA-Ru NPs contains high % of b -sheet, does this affect the cancer cellular uptake? Circular Dichroism of HSA-FA-Ru NPs would be the spectroscopic technique to assess the amount of b-sheet. If indeed your HSA-FA-Ru NPs does contain higher % of b -sheet, would an HSA-FA-Ru NPs with similar HAS secondary structure content be better? These questions need to be addressed.
Page 5, line 165 – labelling and not “abeling”
Page 5, line 168-170 - The sentence “From fluorescence microscopy images 168 (Figure 3) it was observed that both HSA NPs (Figure 3A) and HSA-FA-Ru NPs (Figure 169 3B) formed large aggregates and tended to structure into amyloid-like structures.” confirm that HSA-FA-Ru NPs appears to have much more ThT fluorescence consistent with higher b-sheet content than HSA-NP.
Page 6, line 199 - Not clear what are the concentrations of the red curves. Please add their values in the figures as well and clarify this in the legend.
Page 10, line 343 – Please include EDC and NHS in the list of Abbreviations and what do they stand for.
Page 11, line 401 - Please add the MTS supplier name.
Author Response
Response to Reviewer 1 comments and suggestions
The higher fluorescence of ThT bound to HSA-FA:Ru NPs due to a high HSA b-sheet content should be confirmed by circular dichroism (CD). An increased b-sheet conformation of HSA might affect the cancer cellular uptake of the HSA-FA:Ru NPs reducing their uptake. It is therefore important to assess whether the b-sheet content can also be reduced. The size of the NPs claimed to be about 21nm, yet forming aggregates by fluorescence in the 100 of microns. Controlling the aggregations would certainly improve their cellular uptake.
Answer:
The objective of this study was to synthesize and characterize folic acid-conjugated human serum albumin nanoparticles as targeted carriers for rutin. In a previous study, we investigated and optimized a similar drug delivery system. The novelty of the current work lies in the incorporation of folic acid into the nanoparticle system to enhance cellular uptake.
Regarding the reviewer’s comment about circular dichroism (CD) analysis, we acknowledge its importance and hope that funding for such equipment will become available at our institutions in the near future.
As for the observed aggregation phenomena, it is important to consider the differences in nanoparticle concentration used in the uptake versus aggregation studies. While cell lines were incubated with nanoparticle concentrations up to 2 mg/mL, the aggregation studies—which likely involve alterations in tertiary structure, nucleation, and fibril growth—were conducted at significantly higher concentrations.
To clarify these points, the manuscript has been revised to include the relevant explanations.
Parallel to Voltammetric assay results, it is important to determine the content of FA linked to HSA and assess the reproducibility of HSA-FA:Ru NPs preparation by analysing different batches with UV and CD. The Abs and CD spectral features of FA and Ru do not overlap with those of HSA at wavelengths greater than 300nm (from the literature, both FA and Ru show Abs at about 350nm.
Answer:
Using cyclic voltammetry, the content of FA linked to HSA NP was quantified. Unlike Ru which is predominantly encapsulated within the albumin NPs, FA is bounded on the NP surfaces being accessible for oxidation and accurate quantification.
For these reasons I recommend a major revision for this manuscript.
Here is the list of questions to address.
- Page 1, line 14 – I read “. functionalized with folic acid to promote selective uptake by cancer cells overexpressing folate receptors.” as the folate is covalently attached to the HSA.
Answer:
The HSA-FA:Ru NPs system was designed to be used as a tumor-targeting vector, including carrying a therapeutic agent. In this context, Ru serves as the therapeutic cargo. The covalent conjugation of NPs with folic acid enables targeted delivery by binding to folate receptors of tumor cell membranes. As the cancecer cells express much more folic acid receptors than normal cells, the claim that folic acid bound to NPs promotes selective uptake by cancer cells is correct.
- Page 1, line 18-19 – If I understood correctly that folic acid is covalently link to HSA what about rutin? From “Voltametric measurements confirmed the successful incorporation of both rutin and folic acid within HSA-FA-Ru nanoparticle formulations” the name HSA-FA-Ru would indicate that both FA and Ru are covalently linked to HSA. However, as only FA is covalently linked to HSA as I assume it is, then the nanoparticle should be labelled (HSA-FA):Ru to avoid any confusing and misunderstanding.
Answer:
Your observation is correct - FA binds covalently to HSA (see answer 1), that's why the compound notation is HSA-FA-Ru and not HSA-Ru-FA. Please accept our chosen notation, which is also used in the literature for other compounds.
- Page 1, line 21 – Please add what MTS stands for and MTS to be included in the list of Abbreviations on page 13
Answer:
Thank you for your suggestion. We made the suggested addition as follows:
’’MTS colorimetric test revealed that HSA-FA-Ru NPs significantly reduced the viability of HT-29 cells….’’.
Also, the Materials section has been completed as follows: ’’The MTS (3-(4,5-dimethylthiazol-2-yl)-5-(3-carboxymethoxyphenyl)-2-(4-sulfophenyl)-2H-tetrazolium) kit was also purchased from Thermo Fisher Scientific, Waltham, Massachusetts, USA.’’
- Page 4, line 151 – The sentence “Albumin possesses a tertiary structure characterized by a high content of β-sheet motifs” is not correct. HSA contains a small amount of b-sheet, about 10%. If indeed your HSA-FA-Ru NPs contains high % of b -sheet, does this affect the cancer cellular uptake? Circular Dichroism of HSA-FA-Ru NPs would be the spectroscopic technique to assess the amount of b-sheet. If indeed your HSA-FA-Ru NPs does contain higher % of b -sheet, would an HSA-FA-Ru NPs with similar HAS secondary structure content be better? These questions need to be addressed.
Answer:
Thank you for the observation, you are right! ’’Albumin possesses a secondary structure characterized by 10% content of β-sheet motifs, which are often retained during NPs formation.’’
However, many studies have reported that albumins can form large aggregates resembling amyloid-like structures in vitro. A critical step in amyloid fibril formation is the destabilization of native conformation, typically induced by changes in pH, temperature, ionic strength, and other environmental factors [https://doi.org/10.1016/j.saa.2021.119489; https://doi.org/10.1016/j.ijbiomac.2024.132549]. Since the protocol used in this study avoided such destabilizing conditions—and in light of the stability data presented in the Supporting Information—we investigated the effect of concentration on fibril formation using Thioflavin T as a fluorescence probe for fibrillization. The manuscript was modified.
- Page 5, line 165 – labelling and not “abeling”
Answer:
The manuscript was modified.
’’Thus, HSA-NPs and HSA-FA-Ru NPs were labeled with the fluorescent indicator thioflavin T (ThT), and labeling was performed by incubating NPs of concentration 200 mg/mL with 1 mM thioflavin T in a working volume of 1 mL, followed by centrifugation and washing.’’
- Page 5, line 168-170 - The sentence “From fluorescence microscopy images 168 (Figure 3) it was observed that both HSA NPs (Figure 3A) and HSA-FA-Ru NPs (Figure 169 3B) formed large aggregates and tended to structure into amyloid-like structures.” confirm that HSA-FA-Ru NPs appears to have much more ThT fluorescence consistent with higher b-sheet content than HSA-NP.
Answer:
Indeed, the both NP systems present fibrillation propensity, which indicate a structural modification with an increase in b-sheet. One should be taken into consideration, the level of protein concentration for this kind of aggregation take place. While cell lines were incubated with nanoparticle concentrations up to 2 mg/mL, the aggregation studies—which likely involve alterations in tertiary structure, nucleation, and fibril growth—were conducted at significantly higher concentration, 200 mg/mL.
- Page 6, line 199 - Not clear what are the concentrations of the red curves. Please add their values in the figures as well and clarify this in the legend.
Answer:
Due to their redox-active properties, both rutin and folic acid are suitable for detection and quantification using voltammetric techniques. The standard addition method was employed to determine the unknown concentration of the analyte (A) rutin and B) folic acid. Therefore, the concentrations of red curves have to be determined and this was done by plotting the currents for both standards (A2 and B2). The OX intercept for each plot represents the unknown concentration.
- Page 10, line 343 – Please include EDC and NHS in the list of Abbreviations and what do they stand for.
Answer:
Thank you for the suggestion. EDC/NHS is used to conjugate proteins, peptides or other biomolecules. The list of abbreviations has been completed on your recommendation.
’’The following abbreviations are used in this manuscript:
- Page 11, line 401 - Please add the MTS supplier name.
Answer:
The supplier is specified in the text. See answer to item 3.

Reviewer 2 Report (New Reviewer)
Comments and Suggestions for Authors
1. The manuscript states an HSA solution concentration of 200 mg/mL (Page 10), significantly exceeding typical ranges (usually ≤50 mg/mL). Was the impact of this high concentration on nanoparticle formation validated? Could it lead to aggregation or affect particle size homogeneity?
2. Results 2.2: A free folic acid competition assay was not included: If HT-29 folate receptors are pre-blocked with free FA, is the uptake and cytotoxicity of HSA-FA-Ru NPs inhibited? This is a crucial control experiment to demonstrate FA-mediated targeting specificity.
3. Fig 5: Is the highest concentration tested for cytotoxicity (2 mg/mL NPs) physiologically relevant? Please clarify the rationale for selecting this concentration and discuss its feasibility for in vivo application.
4. Stability was assessed solely by UV absorbance at 280 nm (Page 13). Data on particle size or Zeta potential changes over time are missing. How was interference from aggregation on absorbance readings excluded?
5. Results 2.1: AFM measured HSA-FA-Ru NPs height as 21.7±1.76 nm, while SEM indicated sizes of 50–70 nm (Page 3). The authors attribute this to substrate differences, but further explanation is needed: AFM measures "height" while SEM measures "projected diameter" – are these directly comparable? Could particle collapse during drying contribute to the difference?
6. The text states that 2 mg/mL HSA-FA-Ru NPs reduced HT-29 viability "below 50%" (Page 7), but Fig 5B shows viability at approximately 60% for this concentration. Please verify data consistency
7. The hypothesis suggests Ru release upon intracellular NP degradation by proteases (Page 3), but release kinetics (e.g., in vitro simulation of lysosomal conditions) were not investigated. Was the correlation between release rate and cytotoxicity validated?
8. Keywords are limited to only 3 terms ("cell biocompatibility" is overly broad). Suggest adding relevant terms like "folic acid targeting"
Author Response
Response to Reviewer 2 comments and suggestions
- The manuscript states an HSA solution concentration of 200 mg/mL (Page 10), significantly exceeding typical ranges (usually ≤50 mg/mL). Was the impact of this high concentration on nanoparticle formation validated? Could it lead to aggregation or affect particle size homogeneity?
Answer:
Indeed, a concentration value of 200 mg/mL, for a initial volume of 12 mL, exceeds the usual range used in the literature, but it should be taken into consideration the fact that the final volume of NPs solutions was 100 mL (24 mg mL-1) These conditions were chosen following preliminary optimization studies, [ref coating], which showed a higher efficiency in nanoparticle formation. Additionally, as already mentioned in the manuscript, there was a second experiments directed to the investigation of aggregation where the high concentration of 200 mg/mL was used.
- Results 2.2: A free folic acid competition assay was not included: If HT-29 folate receptors are pre-blocked with free FA, is the uptake and cytotoxicity of HSA-FA-Ru NPs inhibited? This is a crucial control experiment to demonstrate FA-mediated targeting specificity.
Answer:
Thank you for your observation. We agree that a free folic acid competition assay would be a valuable addition to confirm the specificity of folate receptor-mediated uptake of the HSA-FA-Ru nanoparticles. Notably, the culture medium used (DMEM) contains approximately 9 μM folic acid, the same concentration calculated by cyclic voltammetry for 1 mg/mL HSA-FA-Ru NPs. Indeed, an extensive investigation for a broad range of concentrations will be more relevant. This investigation, together with Western Blot and RT-qPCR to asses folate receptors expression, on health and pathogenic cancer cell lines, will be subject of a future work.
- Fig 5: Is the highest concentration tested for cytotoxicity (2 mg/mL NPs) physiologically relevant? Please clarify the rationale for selecting this concentration and discuss its feasibility for in vivo
Answer:
The concentration range was selected experimental and was based on two main considerations: first, to reach a cytotoxic effect on cancer line, and second, to maintain a high viability, even for the upper limits of nanoparticle concentration, in healthy cells. Direct translation of this concentration to the in vivo context requires further analysis. In general, doses used in vivo are adjusted for systemic distribution, volume of administration, and route of delivery, so the effective local concentration may be much lower. We added in the text the following discussion related to the feasibility of applying this concentration in vivo, as well as a clearer justification for the choice of the range tested in vitro, thus:
’’The maximum concentration tested in vitro (2 mg/mL) allows the full spectrum of biological response to be characterized, including effects at high doses. In vivo, the effective concentration at tumor level is significantly influenced by biodistribution, volume of administration and systemic clearance. Although the concentration of 2 mg/mL does not directly reflect a physiological dose, it provides useful information for assessing therapeutic potential. For in vivo applications pharmacokinetic and biodistribution studies will be required.’’
- Stability was assessed solely by UV absorbance at 280 nm (Page 13). Data on particle size or Zeta potential changes over time are missing. How was interference from aggregation on absorbance readings excluded?
Answer:
In our study, absorption at 280 nm was used as a simple and rapid method to monitor the overall stability of nanoparticles, given the presence of albumin in their composition. To reduce the possibility of interference caused by aggregation in the absorbance measurements, samples were visually inspected for signs of opalescence or precipitate and were diluted appropriately before analysis. Measurements were also made immediately after preparation and at regular intervals, and constant absorbance values indicated that the overall integrity of the suspension was maintained.
Accepting these limitations, we consider that the chosen approach was adequate for the purpose of the study and provides a relevant estimate of short-term stability under the conditions tested.
Concerning the reviewer’s comment about DLS analysis, we acknowledge its importance and hope that funding for such equipment will become available at our institutions in the near future.
- Results 2.1: AFM measured HSA-FA-Ru NPs height as 21.7±1.76 nm, while SEM indicated sizes of 50–70 nm (Page 3). The authors attribute this to substrate differences, but further explanation is needed: AFM measures "height" while SEM measures "projected diameter" – are these directly comparable? Could particle collapse during drying contribute to the difference?
Answer:
Thank you for your observation. The difference between AFM and SEM values indeed reflects the different nature of the measured parameters: AFM determines the particle height on a solid surface, while SEM gives the projected diameter. These parameters are not directly comparable. In addition, the drying process can lead to partial collapse of soft particles on the surface of the AFM substrate, resulting in lower height values.
Tre manuscripi wea completede as follows:
‘The sizes obtained by SEM for the two types of NPs were slightly larger than those measured by AFM. This discrepancy may be attributed to: (1) the different types of surfaces on which the NPs were deposited, (2) the distinct nature of the parameters measured by each technique, or (3) sampling-related effects. Nevertheless, all measured sizes were below 100 nm, which supports their intended application for cellular interaction. Although the values obtained by SEM and AFM are not directly comparable, the two techniques provide complementary information regarding particle morphology and structure. Specifically, AFM measures particle height, while SEM determines the projected diameter’’
- The text states that 2 mg/mL HSA-FA-Ru NPs reduced HT-29 viability "below 50%" (Page 7), but Fig 5B shows viability at approximately 60% for this concentration. Please verify data consistency
Answer:
Thank you for your observation. The viability of HT-29 incubated with 2 mg/mL HSA-FA-Ru NPs was a little over 50%. The text was modified acordint to reviewer observation:
By using HSA-FA-Ru NPs, a slightly decrease of viability, but still elevated, was obtained for the L929 line, about 70.66% (Figure 5A (●), for NPs concentrations above 1 mg/mL, whereas higher concentrations (around 2 mg/mL) of NPs induced a decrease of viability, at 54.58% (Figure 5B (●), for the HT-29 cell line.
- The hypothesis suggests Ru release upon intracellular NP degradation by proteases (Page 3), but release kinetics (e.g., in vitrosimulation of lysosomal conditions) were not investigated. Was the correlation between release rate and cytotoxicity validated?
Answer:
Thank you for your observation. The hypothesis of intracellular release was based on the properties of albumin as a proteolytic substrate, as well as the cytotoxic effects observed specifically in tumor cells. The observed dose-dependent cytotoxic effects suggest intracellular availability of the therapeutic agent, but we agree that release kinetics studies would significantly strengthen this conclusion.In the future, we will consider controlled release in lysosomal media to clarify the direct relationship between the release rate of rutin-containing NPs and the observed biological effect.
- Keywords are limited to only 3 terms ("cell biocompatibility" is overly broad). Suggest adding relevant terms like "folic acid targeting"
Answer:
Thank you for this suggestion. The keywords were modified:
‘’human serum albumin nanoparticles; folic acid targeting; rutin delivery system’’

Reviewer 3 Report (New Reviewer)
Comments and Suggestions for Authors
The manuscript “Human serum albumin-based nanoparticles for targeted intracellular drug delivery” is a study about two HAS based nanoparticles, one of them loaded with folic acid and Rutin. The manuscript is interesting and well written; however, it needs some corrections before it can be ready to be published. Here are some comments and questions:
1 Section “Preparation of HSA NPs” and “Preparation of HSA-FA-Ru loaded nanoparticles”, please indicate the volumes of the HSA, glucose and FA solutions.
2 Figure 2 caption indicates nanoparticles were deposited in “Au-coated Si”, but in line 131 the author indicate they were deposited on Au-coated glass, please correct where is appropriated.
3 AFM results. I don’t see the results obtained by AFM as sound as they could be. If the authors were trying to visualize nanoparticles of sizes around tens of nanometers, why did they scan a 5x5 micrometer area? I think a better prepared sample (more diluted) could help to see less aggregates and asses the size of the nanoparticles in a more precise way. The size for the HSA-FA-Ru is reported in the text as 21.70±1.76 nm, but in figure 1F it is labeled as 21.70±7.76 nm, please correct.
4 SEM results. Authors wrote in lines 138-141 “The sizes obtained by SEM for the two types of NPs were slightly larger than those obtained by AFM, which may be due to the different type of surface on which they were deposited, but they are below 100 nm, which makes the purpose for which they were prepared, to be applied to cells, achievable.” but AFM could be performed in the Au coated glass substrates, this will reduce the differences in sizes measured by the two techniques. I firmly believe that the size characterization must be bettered since even with the two techniques applied, as a reader, I don’t know if the particles are in the 10-20 nm range or in the 70-100 nm one. And the whole point of nanotechnology is that size do matter, so it is important to define what are the characteristics of the materials they are evaluating in the cells.
5 Authors must perform a more complete characterization of the HAS and HAS-FA-Ru nanoparticles. FT-IR, XPS, UV-Vis, DLS and Z potential will give a better and more complete insight of the NPs physical and chemical properties. Check: https://doi.org/10.1016/j.ijbiomac.2019.10.013 https://doi.org/10.1186/1556-276X-9-343 https://doi.org/10.1021/acsami.6b00857
6 I suggest that the authors correct error bars colors in figure 5b to match the colors of the symbols in the plots.
7 The SEM images for the uptake of HT-29 cells are missing.
8 SEM images in figure 6 certainly shows that “the interaction with nanoparticles does not induce morphological changes” but these image, with this resolution, cannot be used to affirm that “NPs tend to concentrate on the cell surface” or “HSA-FA-Ru NPs are predominantly found around and on the cell surface”, in fact, these HSA-FA-Ru NPs are shown as they don interact with the L929 cell.
9 Figure C2 should be changed for one that depicts more than one cell.
10 Conclusion lines 417-419 “This result, together with the time stability of the NPs, provided the basis for in vitro studies monitoring the effect of these NPs on L929 and Ht-29 cells.” The manuscript does not present any study about the “time stability of the NPs”. Please explain this
11 In conclusions lines 424-427 the authors wrote: “Fluorescence microscopy showed that HSA NPs and HSA-FA-Ru NPs do not affect the nucleus and cytoskeleton of L929 and HT-29 cells, but for HT-29 cells a tendency for NPs to agglomerate on the cell surface was observed, and the effect was more evident for HSA-FA-Ru NPs.” But they also wrote in lines 416-417 “Therefore, these NPs represent a valuable approach for the penetration of the membrane and for cellular target.” This is contradictory, please correct.
12 It would be great to see the equivalents of free Rutin and folic acid as controls in the cell viability experiments, by this the authors should be able to demonstrate that de HAS NPs really act as carriers.
13 This is more of a personal opinion, not a criticize to the work, I think that it could be interesting to see how hypothetical HSA-FA and HSA-Ru nanoparticles would perform in the cell internalization and viability.
Author Response
Response to Reviewer 3 comments and suggestions
The manuscript “Human serum albumin-based nanoparticles for targeted intracellular drug delivery” is a study about two HAS based nanoparticles, one of them loaded with folic acid and Rutin. The manuscript is interesting and well written; however, it needs some corrections before it can be ready to be published. Here are some comments and questions:
1 Section “Preparation of HSA NPs” and “Preparation of HSA-FA-Ru loaded nanoparticles”, please indicate the volumes of the HSA, glucose and FA solutions.
Answer:
The manuscript was modified according to reviewer suggestions.
2 Figure 2 caption indicates nanoparticles were deposited in “Au-coated Si”, but in line 131 the author indicate they were deposited on Au-coated glass, please correct where is appropriated.
Answer:
The text has been corrected. Gold coated glass was used.
3 AFM results. I don’t see the results obtained by AFM as sound as they could be. If the authors were trying to visualize nanoparticles of sizes around tens of nanometers, why did they scan a 5x5 micrometer area? I think a better prepared sample (more diluted) could help to see less aggregates and asses the size of the nanoparticles in a more precise way. The size for the HSA-FA-Ru is reported in the text as 21.70±1.76 nm, but in figure 1F it is labeled as 21.70±7.76 nm, please correct.
Answer:
Thank you for your observation. We have corrected the text to ensure consistency between the reported nanoparticle sizes in the figure and the main text. The revised sentence now reads:
“…while HSA-FA-Ru particles range in the interval 21.70 ± 7.76 nm.” Regarding the AFM analysis, we appreciate your valuable feedback. We agree that the 5 × 5 µm scan area is relatively large for assessing nanoparticles in the tens of nm size range. Our intention was to evaluate the overall surface coverage and detect potential aggregation, rather than to resolve individual particles. Considering the limitations encountered with AFM in this context, SEM was employed, as a complementary technique, to further investigate nanoparticle morphology and distribution on the substrate.
4 SEM results. Authors wrote in lines 138-141 “The sizes obtained by SEM for the two types of NPs were slightly larger than those obtained by AFM, which may be due to the different type of surface on which they were deposited, but they are below 100 nm, which makes the purpose for which they were prepared, to be applied to cells, achievable.” but AFM could be performed in the Au coated glass substrates, this will reduce the differences in sizes measured by the two techniques. I firmly believe that the size characterization must be bettered since even with the two techniques applied, as a reader, I don’t know if the particles are in the 10-20 nm range or in the 70-100 nm one. And the whole point of nanotechnology is that size do matter, so it is important to define what are the characteristics of the materials they are evaluating in the cells.
Answer:
Thank you for your observation. We agree that accurate size characterisation is essential. Experimentally, the exact same substrate for AFM and SEM can’t be used for technical reasons: 1) AFM requires a smooth and non-metallic substrate (e.g. mica, glass, oxidized silicon, HOPG), so as not to interfere with the probe. 2) SEM performed on biological samples needs a conductive substrate (e.g. doped silicon, metal, gold-plated glass) to avoid charge build-up.
Indeed, the use of a common substrate, such as gold-coated extremely flat surface would allow more accurate correlation between AFM and SEM. In the future, we will keep this aspect in mind.
To be better explained, the manuscript was modified:
“The sizes obtained by SEM for the two types of NPs were slightly larger than those measured by AFM. This discrepancy may be attributed to: (1) the different types of surfaces on which the NPs were deposited, (2) the distinct nature of the parameters measured by each technique, or (3) sampling-related effects. Nevertheless, all measured sizes were below 100 nm, which supports their intended application for cellular interaction. Although the values obtained by SEM and AFM are not directly comparable, the two techniques provide complementary information regarding particle morphology and structure. Specifically, AFM measures particle height, while SEM determines the projected diameter.”
5 Authors must perform a more complete characterization of the HAS and HAS-FA-Ru nanoparticles. FT-IR, XPS, UV-Vis, DLS and Z potential will give a better and more complete insight of the NPs physical and chemical properties. Check: https://doi.org/10.1016/j.ijbiomac.2019.10.013 https://doi.org/10.1186/1556-276X-9-343 https://doi.org/10.1021/acsami.6b00857
Answer:
Thank you for your suggestions. In this study, the application of other characterization techniques was not possible due to equipment availability and specific experimental conditions.
6 I suggest that the authors correct error bars colors in figure 5b to match the colors of the symbols in the plots.
Answer:
The error bars were modified.
7 The SEM images for the uptake of HT-29 cells are missing.
Answer:
We thank the reviewer for pointing this out. Indeed, the SEM images recorded for HT-29 are missing. In the revised manuscript, the images were included and the description of the results, as well as the legend, were modified.
8 SEM images in figure 6 certainly shows that “the interaction with nanoparticles does not induce morphological changes” but these image, with this resolution, cannot be used to affirm that “NPs tend to concentrate on the cell surface” or “HSA-FA-Ru NPs are predominantly found around and on the cell surface”, in fact, these HSA-FA-Ru NPs are shown as they don interact with the L929 cell.
Answer:
The manuscript was modified according to the reviewer observation and in agreement with the novel images included (see point 7)
9 Figure C2 should be changed for one that depicts more than one cell.
Answer:
Supposing that the reviewer is referee to Fig. 7C2, it should be noted that there is more than one cell, it is a HT-29 cluster. Indeed, the clusters are usually larger, and contains more cells (see the cluster bottom right control (A2). Incubating with both NP systems, a decrease of clusters number and size was observed, with a highest decrease after incubation with HSA-FA-Ru NPs.
10 Conclusion lines 417-419 “This result, together with the time stability of the NPs, provided the basis for in vitro studies monitoring the effect of these NPs on L929 and Ht-29 cells.” The manuscript does not present any study about the “time stability of the NPs”. Please explain this
Answer:
The time stability of nanoparticles is provided in the Supplementary information.
11 In conclusions lines 424-427 the authors wrote: “Fluorescence microscopy showed that HSA NPs and HSA-FA-Ru NPs do not affect the nucleus and cytoskeleton of L929 and HT-29 cells, but for HT-29 cells a tendency for NPs to agglomerate on the cell surface was observed, and the effect was more evident for HSA-FA-Ru NPs.” But they also wrote in lines 416-417 “Therefore, these NPs represent a valuable approach for the penetration of the membrane and for cellular target.” This is contradictory, please correct.
Answer:
Thank you for your pertinent observation. We have corrected the contradiction. Indeed, the fluorescence images showed the tendency of nanoparticles to agglomerate at the surface of HT-29 cells, without providing clear evidence of membrane penetration. We have reworded the passage to correctly reflect these observations. As follows:
’’ Fluorescence microscopy showed that HSA NPs and HSA-FA-Ru NPs do not affect the nucleus and cytoskeleton of L929 and HT-29 cells, but for HT-29 cells a tendency of clusters shrinking was observed, and the effect was more evident for HSA-FA-Ru NPs.’’
12 It would be great to see the equivalents of free Rutin and folic acid as controls in the cell viability experiments, by this the authors should be able to demonstrate that de HAS NPs really act as carriers.
Answer:
Thank you very much for your thoughtful suggestion. We recognize the importance of these controls for FA and Ru to clearly demonstrate the role of HSA nanoparticles as delivery vectors.. However, it's important to highlight that free rutin, due to its high polarity and relatively large molecular weight, has very limited ability to cross the cell membrane on its own. This well-documented limitation was one of the key motivations behind using a nanoparticle-based delivery approach. We performed extensive characterizations of the nanoparticles and demonstrated a significant increase in therapeutic efficacy compared to free molecules, based on literature data and our cell internalization and viability assays. Notably, the culture medium used (DMEM) contains approximately 9 μM folic acid, the same concentration calculated by cyclic voltammetry for 1 mg/mL HSA-FA-Ru NPs. Indeed, an extensive investigation for a broad range of concentrations will be more relevant. This investigation, together with Western Blot and RT-qPCR to asses folate receptors expression, on health and pathogenic cancer cell lines, will be subject of a future work. We believe that, in its current form, the work provides valuable and robust information and the inclusion of controls will be considered in future studies.
13 This is more of a personal opinion, not a criticize to the work, I think that it could be interesting to see how hypothetical HSA-FA and HSA-Ru nanoparticles would perform in the cell internalization and viability.
Answer:
Thank you for the suggestion! We agree that evaluation of HSA-FA and HSA-Ru NPs would be a modality to better understand the individual role of each component in cell internalization and impact on cell viability.
For the future, it would be an interesting idea for a series of comparative experiments, in which HSA-FA NPs and HSA-Ru NPs are synthesized and characterized independently, and then tested on the same cell lines (e.g. by confocal microscopy and flow cytometry). This could yield interesting data on how efficient internalization is. Cell viability assays (MTS or MTT) could also reveal possible differences in cytotoxic effects.
We believe this would be a good way to clarify mechanisms of action and potential synergies between folic acid and Ru in NPs, providing a solid basis for the development of more effective targeted therapies.

Reviewer 4 Report (New Reviewer)
Comments and Suggestions for Authors
This study designed and characterized a folate (FA)-targeted human serum albumin (HSA) nanoparticle (HSA-FA-Ru NPs) for the delivery of the flavonoid compound rutin (Ru), which exhibits anticancer activity. The study systematically evaluated the morphology, particle size, drug loading capacity, cellular uptake, and selective toxicity of this nanoparticle system. The experimental design was well-structured, the data were comprehensive, and the conclusions were clear, demonstrating a certain degree of innovation and application potential. The following are specific review comments:
1. DLS data is required to verify the hydrodynamic size and dispersion of nanoparticles.. 2. Pay attention to the format of your paper. 3. Lack of data on drug release kinetics (such as release curves under different pH conditions). 4. Some figures have low resolution. We recommend providing higher resolution versions. 5. Recommend deleting or merging some of the lengthy background descriptions. 6. Authors should standardize the format of references. 7. Zeta potential affects the colloidal stability and cellular uptake efficiency of nanoparticles, but studies have not provided relevant data. 8. The voltammometric method only detected part of the Ru and FA loading, but did not calculate the overall encapsulation efficiency (EE%) and drug loading (DL%). 9. Although FA receptor-mediated targeting is assumed, it has not been verified by competitive experiments (such as free FA blockade) or receptor expression detection (such as Western blot/flow cytometry).
Authors need to resubmit the manuscript after additional experiments, improved data analysis and discussion.
Comments on the Quality of English Language- The authors should carefully check the manuscript for typos.
- The authors should use a more scientific language to present their findings.
Author Response
Response to Reviewer 4 comments and suggestions
This study designed and characterized a folate (FA)-targeted human serum albumin (HSA) nanoparticle (HSA-FA-Ru NPs) for the delivery of the flavonoid compound rutin (Ru), which exhibits anticancer activity. The study systematically evaluated the morphology, particle size, drug loading capacity, cellular uptake, and selective toxicity of this nanoparticle system. The experimental design was well-structured, the data were comprehensive, and the conclusions were clear, demonstrating a certain degree of innovation and application potential. The following are specific review comments:
- DLS data is required to verify the hydrodynamic size and dispersion of nanoparticles..
Answer:
We acknowledge the importance of DLS analysis and hope that funding for such equipment will become available at our institutions in the near future.
- Pay attention to the format of your paper.
Answer:
The format agrees with IJMS template.
- Lack of data on drug release kinetics (such as release curves under different pH conditions).
Answer:
The objective of this study was to synthesize and characterize folic acid-conjugated human serum albumin nanoparticles as targeted carriers for rutin. In a previous study, we investigated and optimized a similar drug delivery system. The novelty of the current work lies in the incorporation of folic acid into the nanoparticle system to enhance cellular uptake. For the future, it would be an interesting idea for a series of comparative experiments, in which HSA-FA NPs and HSA-Ru NPs are synthesized and characterized independently, and then tested on the same cell lines (e.g. by confocal microscopy and flow cytometry). Also, drug release under different condition will be investigated. This could yield interesting data on how efficient internalization is. Cell viability assays (MTS or MTT) could also reveal possible differences in cytotoxic effects.
We believe this would be a good way to clarify mechanisms of action and potential synergies between folic acid and Ru in NPs, providing a solid basis for the development of more effective targeted therapies.
- Some figures have low resolution. We recommend providing higher resolution versions.
Answer:
The low resolution may be due to the pdf conversion. We provided high quality figures.
- Recommend deleting or merging some of the lengthy background descriptions.
Answer:
The manuscript was modified.
- Authors should standardize the format of references.
Answer:
The references are in agreement with IJMS format
- Zeta potential affects the colloidal stability and cellular uptake efficiency of nanoparticles, but studies have not provided relevant data.
Answer:
Although these measurements were not included in the current version due to technical constraints at the time, we recognize their importance and plan to incorporate them in future studies to give a more complete picture of the system’s physicochemical properties and biological interactions.
- The voltammometric method only detected part of the Ru and FA loading, but did not calculate the overall encapsulation efficiency (EE%) and drug loading (DL%).
Answer:
See the answer to point 3
- Although FA receptor-mediated targeting is assumed, it has not been verified by competitive experiments (such as free FA blockade) or receptor expression detection (such as Western blot/flow cytometry).
Answer:
Thank you for your observation. We agree that a free folic acid competition assay would be a valuable addition to confirm the specificity of folate receptor-mediated uptake of the HSA-FA-Ru nanoparticles. Notably, the culture medium used (DMEM) contains approximately 9 μM folic acid, the same concentration calculated by cyclic voltammetry for 1 mg/mL HSA-FA-Ru NPs. Indeed, an extensive investigation for a broad range of concentrations will be more relevant. This investigation, together with Western Blot and RT-qPCR to asses folate receptors expression, on health and pathogenic cancer cell lines, will be subject of a future work.
Authors need to resubmit the manuscript after additional experiments, improved data analysis and discussion.
Answer:
The manuscript was improved

Reviewer 5 Report (New Reviewer)
Comments and Suggestions for Authors
The article compares the effects of synthesized nanoparticles (NPs) of pure human serum albumin (HSA) and nanoparticles of HSA with rutin (Ru) directed by folic acid (FA) (HSA-FA-Ru-NPs) on fibroblast cell lines L929 and human adenocarcinoma cell line HT-29.
The main results are shown in Figure 5, where the viability of control L929 cells decreased from approximately 90% for neutral HSA NPs to 70% for the proposed HSA-FA-Ru NPs (Fig. 5A), while the action of HSA-FA-Ru NPs caused a decrease in the viability of HT-29 cells to approximately 57% (Fig. 5B).
To improve the manuscript, the following aspects need to be clarified:
1. To convince the reader that the additional 13% suppression of the HT-29 adenocarcinoma cell line is due to the FA-targeted delivery of Ru-containing HAS nanoparticles, it is necessary to compare the effects of 4 types of nanoparticles on both types of cells, namely, nanoparticles based on pure HSA, HSA-FA, HAS-Ru, and HSA-FA-Ru. This may reveal a separate contribution to cell suppression while FA directed of HSA nanoparticles to the HT-29 calls or simply the effect of Ru within HAS NPs on cells as a therapeutic drug.
2. In the introduction, the authors cite the excellent properties of HSA nanoparticles such as “biodegradability, biocompatibility, low toxicity.” How does this fit with the inhibition of both cell types by pure HSA nanoparticles up to 90% in Figures 5A, B?
3. The study of protein particles using AFM and SEM is not very informative, since it strongly depends on the methods of preparation of samples on different substrates, which is often accompanied by particle aggregation. In particular, Fig. 1A shows large clusters of NPs of pure HSA, which visually create the impression of being much larger than the HSA-FA-Ru obtained particles (Fig. 1B), which is inconsistent with the image processing in Fig. 1E,F. Therefore, it would be worthwhile to study the sizes of NPs directly in solution using Nanoparticle Tracking Analysis (NTA) or Dynamic Light Scattering (DLS).
4. Some typos need to be corrected. For example, in line 245, "HSA-FA-Ru NPs" describes Figure 5A, while in the caption to this Figure, line 254, it says "HSA-Ru NPs".
Author Response
Response to Reviewer 5 comments and suggestions
The article compares the effects of synthesized nanoparticles (NPs) of pure human serum albumin (HSA) and nanoparticles of HSA with rutin (Ru) directed by folic acid (FA) (HSA-FA-Ru-NPs) on fibroblast cell lines L929 and human adenocarcinoma cell line HT-29.
The main results are shown in Figure 5, where the viability of control L929 cells decreased from approximately 90% for neutral HSA NPs to 70% for the proposed HSA-FA-Ru NPs (Fig. 5A), while the action of HSA-FA-Ru NPs caused a decrease in the viability of HT-29 cells to approximately 57% (Fig. 5B).
To improve the manuscript, the following aspects need to be clarified:
- To convince the reader that the additional 13% suppression of the HT-29 adenocarcinoma cell line is due to the FA-targeted delivery of Ru-containing HAS nanoparticles, it is necessary to compare the effects of 4 types of nanoparticles on both types of cells, namely, nanoparticles based on pure HSA, HSA-FA, HAS-Ru, and HSA-FA-Ru. This may reveal a separate contribution to cell suppression while FA directed of HSA nanoparticles to the HT-29 calls or simply the effect of Ru within HAS NPs on cells as a therapeutic drug.
Answer:
Indeed, a full comparison of HSA, HSA-FA, HSA-Ru and HSA-FA-Ru nanoparticles, tested on both FA receptor and non-FA receptor cell lines, would provide important clarifications regarding the mechanism of the observed effect. At this stage, our study focused on the evaluation of the therapeutic efficacy of the HSA-FA-Ru formulation, and the mentioned additional data were not available. We revised the manuscript to emphasize this limitation and added a mention in the discussion section stating that:
''One perspective of these studies, will be to compare these results with those obtained for testing the effect of HSA, HSA-FA and HSA-Ru NPs at the cellular level, to determine the specific contribution of FA targeting towards the cytotoxic effect of the Ru complex.''
- In the introduction, the authors cite the excellent properties of HSA nanoparticles such as “biodegradability, biocompatibility, low toxicity.” How does this fit with the inhibition of both cell types by pure HSA nanoparticles up to 90% in Figures 5A, B?
Answer:
Thank you for your observation. Although HSA is described in the literature as biocompatible and of low toxicity, in our experiments we observed an important inhibition of cell proliferation, This effect may be due on the one hand to the concentration of HSA in our samples, on the other hand to the physical interaction of the particles with the cell surface. Also, it shold be notted that 90% viability represents a high biocompatibility and low toxicity. We introduced this explanation in the discussion of the paper to reflect this observation, as follows:
“Altrouh HSA shows properties as biodegradability, biocompatibility, low toxicity, our observations suggest that under certain conditions, HSA NPs may have notable biological effects that must be clarified in future investigation.”
- The study of protein particles using AFM and SEM is not very informative, since it strongly depends on the methods of preparation of samples on different substrates, which is often accompanied by particle aggregation. In particular, Fig. 1A shows large clusters of NPs of pure HSA, which visually create the impression of being much larger than the HSA-FA-Ru obtained particles (Fig. 1B), which is inconsistent with the image processing in Fig. 1E,F. Therefore, it would be worthwhile to study the sizes of NPs directly in solution using Nanoparticle Tracking Analysis (NTA) or Dynamic Light Scattering (DLS).
Answer:
Thank you for your suggestions. In this study, the application of other characterization techniques was not possible due to equipment availability and specific experimental conditions.
- Some typos need to be corrected. For example, in line 245, "HSA-FA-Ru NPs" describes Figure 5A, while in the caption to this Figure, line 254, it says "HSA-Ru NPs".
Answer:
The typo has been corrected – the term “HSA-FA-Ru NPs” in line 245 is now consistent with the caption of Figure 5A, where it appears as “HSA-FA-Ru NPs”.

Round 2
Reviewer 1 Report (New Reviewer)
Comments and Suggestions for Authors
My comments to this response are written below in red
Response to Reviewer 1 comments and suggestions
The higher fluorescence of ThT bound to HSA-FA:Ru NPs due to a high HSA b-sheet content should be confirmed by circular dichroism (CD). An increased b-sheet conformation of HSA might affect the cancer cellular uptake of the HSA-FA:Ru NPs reducing their uptake. It is therefore important to assess whether the b-sheet content can also be reduced. The size of the NPs claimed to be about 21nm, yet forming aggregates by fluorescence in the 100 of microns. Controlling the aggregations would certainly improve their cellular uptake.
Answer:
The objective of this study was to synthesize and characterize folic acid-conjugated human serum albumin nanoparticles as targeted carriers for rutin. In a previous study, we investigated and optimized a similar drug delivery system. The novelty of the current work lies in the incorporation of folic acid into the nanoparticle system to enhance cellular uptake.
Regarding the reviewer’s comment about circular dichroism (CD) analysis, we acknowledge its importance and hope that funding for such equipment will become available at our institutions in the near future.
In this case CD is an essential measurement to be done as it measures directly the folding of the protein without having to label the protein, which could affect the folding. You could have applied for CD beamtime or search for collaborations with other research groups to include the CD spectrum and its secondary structure content rather quickly.
However, the fact that the (HSA-FA):Ru did reduce the cells viability is the prove this system can work. Any future work to optimise this delivery system must use the CD measurements making sure the HSA folding is fully retained or as much as possible.
As for the observed aggregation phenomena, it is important to consider the differences in nanoparticle concentration used in the uptake versus aggregation studies. While cell lines were incubated with nanoparticle concentrations up to 2 mg/mL, the aggregation studies—which likely involve alterations in tertiary structure, nucleation, and fibril growth—were conducted at significantly higher concentrations.
To clarify these points, the manuscript has been revised to include the relevant explanations.
Parallel to Voltammetric assay results, it is important to determine the content of FA linked to HSA and assess the reproducibility of HSA-FA:Ru NPs preparation by analysing different batches with UV and CD. The Abs and CD spectral features of FA and Ru do not overlap with those of HSA at wavelengths greater than 300nm (from the literature, both FA and Ru show Abs at about 350nm.
Answer:
Using cyclic voltammetry, the content of FA linked to HSA NP was quantified. Unlike Ru which is predominantly encapsulated within the albumin NPs, FA is bounded on the NP surfaces being accessible for oxidation and accurate quantification.
For these reasons I recommend a major revision for this manuscript.
Here is the list of questions to address.
- Page 1, line 14 – I read “. functionalized with folic acid to promote selective uptake by cancer cells overexpressing folate receptors.” as the folate is covalently attached to the HSA.
Answer:
The HSA-FA:Ru NPs system was designed to be used as a tumor-targeting vector, including carrying a therapeutic agent. In this context, Ru serves as the therapeutic cargo. The covalent conjugation of NPs with folic acid enables targeted delivery by binding to folate receptors of tumor cell membranes. As the cancer cells express much more folic acid receptors than normal cells, the claim that folic acid bound to NPs promotes selective uptake by cancer cells is correct.
- Page 1, line 18-19 – If I understood correctly that folic acid is covalently link to HSA what about rutin? From “Voltametric measurements confirmed the successful incorporation of both rutin and folic acid within HSA-FA-Ru nanoparticle formulations” the name HSA-FA-Ru would indicate that both FA and Ru are covalently linked to HSA. However, as only FA is covalently linked to HSA as I assume it is, then the nanoparticle should be labelled (HSA-FA):Ru to avoid any confusing and misunderstanding.
Answer:
Your observation is correct - FA binds covalently to HSA (see answer 1), that's why the compound notation is HSA-FA-Ru and not HSA-Ru-FA. Please accept our chosen notation, which is also used in the literature for other compounds.
I insist you should use (HSA-FA):Ru instead of HAS-FA-Ru as it is still confusing rather than just on line 257.
- Page 1, line 21 – Please add what MTS stands for and MTS to be included in the list of Abbreviations on page 13
Answer:
Thank you for your suggestion. We made the suggested addition as follows:
’’MTS colorimetric test revealed that HSA-FA-Ru NPs significantly reduced the viability of HT-29 cells….’’.
Also, the Materials section has been completed as follows: ’’The MTS (3-(4,5-dimethylthiazol-2-yl)-5-(3-carboxymethoxyphenyl)-2-(4-sulfophenyl)-2H-tetrazolium) kit was also purchased from Thermo Fisher Scientific, Waltham, Massachusetts, USA.’’
- Page 4, line 151 – The sentence “Albumin possesses a tertiary structure characterized by a high content of β-sheet motifs” is not correct. HSA contains a small amount of b-sheet, about 10%. If indeed your HSA-FA-Ru NPs contains high % of b -sheet, does this affect the cancer cellular uptake? Circular Dichroism of HSA-FA-Ru NPs would be the spectroscopic technique to assess the amount of b-sheet. If indeed your HSA-FA-Ru NPs does contain higher % of b -sheet, would an HSA-FA-Ru NPs with similar HSA secondary structure content be better? These questions need to be addressed.
Answer:
Thank you for the observation, you are right! ’’Albumin possesses a secondary structure characterized by 10% content of β-sheet motifs, which are often retained during NPs formation.’’
However, many studies have reported that albumins can form large aggregates resembling amyloid-like structures in vitro. A critical step in amyloid fibril formation is the destabilization of native conformation, typically induced by changes in pH, temperature, ionic strength, and other environmental factors [https://doi.org/10.1016/j.saa.2021.119489; https://doi.org/10.1016/j.ijbiomac.2024.132549]. Since the protocol used in this study avoided such destabilizing conditions—and in light of the stability data presented in the Supporting Information—we investigated the effect of concentration on fibril formation using Thioflavin T as a fluorescence probe for fibrillization. The manuscript was modified.
- Page 5, line 165 – labelling and not “abeling”
Answer:
The manuscript was modified.
’’Thus, HSA-NPs and HSA-FA-Ru NPs were labeled with the fluorescent indicator thioflavin T (ThT), and labeling was performed by incubating NPs of concentration 200 mg/mL with 1 mM thioflavin T in a working volume of 1 mL, followed by centrifugation and washing.’’
- Page 5, line 168-170 - The sentence “From fluorescence microscopy images 168 (Figure 3) it was observed that both HSA NPs (Figure 3A) and HSA-FA-Ru NPs (Figure 169 3B) formed large aggregates and tended to structure into amyloid-like structures.” confirm that HSA-FA-Ru NPs appears to have much more ThT fluorescence consistent with higher b-sheet content than HSA-NP.
Answer:
Indeed, the both NP systems present fibrillation propensity, which indicate a structural modification with an increase in b-sheet. One should be taken into consideration, the level of protein concentration for this kind of aggregation take place. While cell lines were incubated with nanoparticle concentrations up to 2 mg/mL, the aggregation studies—which likely involve alterations in tertiary structure, nucleation, and fibril growth—were conducted at significantly higher concentration, 200 mg/mL.
- Page 6, line 199 - Not clear what are the concentrations of the red curves. Please add their values in the figures as well and clarify this in the legend.
Answer:
Due to their redox-active properties, both rutin and folic acid are suitable for detection and quantification using voltammetric techniques. The standard addition method was employed to determine the unknown concentration of the analyte (A) rutin and B) folic acid. Therefore, the concentrations of red curves have to be determined and this was done by plotting the currents for both standards (A2 and B2). The OX intercept for each plot represents the unknown concentration.
This answer should be included in the Materials and Methods under Voltametric measurements and the electrochemical cell section.
- Page 10, line 343 – Please include EDC and NHS in the list of Abbreviations and what do they stand for.
Answer:
Thank you for the suggestion. EDC/NHS is used to conjugate proteins, peptides or other biomolecules. The list of abbreviations has been completed on your recommendation.
’’The following abbreviations are used in this manuscript:
- Page 11, line 401 - Please add the MTS supplier name.
Answer:
The supplier is specified in the text. See answer to item 3.
Author Response
Comment 1: In this case CD is an essential measurement to be done as it measures directly the folding of the protein without having to label the protein, which could affect the folding. You could have applied for CD beamtime or search for collaborations with other research groups to include the CD spectrum and its secondary structure content rather quickly.
However, the fact that the (HSA-FA):Ru did reduce the cells viability is the prove this system can work. Any future work to optimise this delivery system must use the CD measurements making sure the HSA folding is fully retained or as much as possible.
Answer 1:
Thank you for your valuable revision and for understanding the limitations we faced regarding CD measurements at this moment. However, considering your recommendation, we will include circular dichroism analysis in any future investigations on this topic.
Comment 2: I insist you should use (HSA-FA):Ru instead of HAS-FA-Ru as it is still confusing rather than just on line 257.
Answer 2:
The manuscript was modified according to reviewer suggestion.
Comment 3: This answer should be included in the Materials and Methods under Voltametric measurements and the electrochemical cell section.
Answer 3:
The manuscript was modified. A new paragraph was included the Materials and Methods under Voltametric measurements and the electrochemical cell section: “Due to their redox-active properties, both rutin and folic acid are suitable for de-tection and quantification using voltammetric techniques. The standard addition method was employed to determine the unknown concentration of the analyte Fig. 4. (A) rutin and (B) folic acid. Therefore, the concentrations of red curves have to be de-termined and this was done by plotting the currents for both standards (A2 and B2). The OX intercept for each plot represents the unknown concentration.”
Reviewer 2 Report (New Reviewer)
Comments and Suggestions for Authors
Authors has revised the manuscipt and it could be accepted.
Author Response
Comments and Suggestions for Authors
Authors has revised the manuscipt and it could be accepted.
Thank you for your evaluation.
Reviewer 3 Report (New Reviewer)
Comments and Suggestions for Authors
After carefully revising the version 2 of the manuscript “Human serum albumin-based nanoparticles for targeted intracellular drug delivery” I cannot recommend it publication in Int. J. Mol. Sci. mainly because the characterization of the nanomaterials is incomplete.
C1 Abot the size of the nanoparticles: I my past revision I asked about the discrepancies in the size of the nanoparticles reported by the two techniques, SEM and AFM; the authors responded in lines 144-147 that “Although the values obtained by SEM and AFM are not directly comparable, the two techniques provide complementary information regarding particle morphology and structure. Specifically, AFM measures particle height, while SEM determines the projected diameter.” This mean that they expect that the NPs are not spherical? Because the height and the projected diameter will be in the same ranges for spherical NPS. Then in lines 136-139 “For the HSA NPs, the SEM images obtained (Figure 2A) showed a uniform distribution, these NPs having small sizes of about 50 - 70 nm. Similarly, HSA-FA-Ru NPs (Figure 2B) showed sizes in the tens of nanometers range, while forming aggregates in the hundreds of micrometers range.” But no distribution is shown for these measurements. In lines 127-130 the authors mentioned that “According to literature, HSA NPs dimensions are highly dependent on the processing conditions, such as pH, concentration, temperature or desolvation agent [27]. It is expected that small-sized nano-particles enhance tissue penetration, making HSA and HSA-FA-Ru NPs suitable for targeted delivery of Ru into cancer cells [28, 29].” If the authors recognize the importance of the size for the NPs to act as Ru carriers, how can be acceptable to report a nanomaterial whose size is unknown? I hold my criticism in this part of the manuscript as the size was not measured correctly.
C2 in the past revision I suggested that a more complete characterization must be performed on the HAS and HAS-FA-Ru nanoparticles with techniques such as FT-IR, XPS, UV-Vis, DLS and Z potential will give a better and more complete insight of the NPs physical and chemical properties. No more characterizations were performed to the NPs, and the ones performed are shallow (SEM and AFM). This corresponds to a low-quality scientific report that is not enough to be published in Int. J. Mol. Sci.
Author Response
C1 Abot the size of the nanoparticles: I my past revision I asked about the discrepancies in the size of the nanoparticles reported by the two techniques, SEM and AFM; the authors responded in lines 144-147 that “Although the values obtained by SEM and AFM are not directly comparable, the two techniques provide complementary information regarding particle morphology and structure. Specifically, AFM measures particle height, while SEM determines the projected diameter.” This mean that they expect that the NPs are not spherical? Because the height and the projected diameter will be in the same ranges for spherical NPS. Then in lines 136-139 “For the HSA NPs, the SEM images obtained (Figure 2A) showed a uniform distribution, these NPs having small sizes of about 50 - 70 nm. Similarly, HSA-FA-Ru NPs (Figure 2B) showed sizes in the tens of nanometers range, while forming aggregates in the hundreds of micrometers range.” But no distribution is shown for these measurements. In lines 127-130 the authors mentioned that “According to literature, HSA NPs dimensions are highly dependent on the processing conditions, such as pH, concentration, temperature or desolvation agent [27]. It is expected that small-sized nano-particles enhance tissue penetration, making HSA and HSA-FA-Ru NPs suitable for targeted delivery of Ru into cancer cells [28, 29].” If the authors recognize the importance of the size for the NPs to act as Ru carriers, how can be acceptable to report a nanomaterial whose size is unknown? I hold my criticism in this part of the manuscript as the size was not measured correctly.
Answer:
As previously stated, the primary objective of our work was to demonstrate the potential of our HSA-FA-Ru nanoparticle system as an efficient and selective intracellular delivery platform for rutin, rather than to provide an exhaustive physicochemical characterization of the nanomaterials. In our revision, we have clarified the experimental limitations of morphological characterization.
C2 in the past revision I suggested that a more complete characterization must be performed on the HAS and HAS-FA-Ru nanoparticles with techniques such as FT-IR, XPS, UV-Vis, DLS and Z potential will give a better and more complete insight of the NPs physical and chemical properties. No more characterizations were performed to the NPs, and the ones performed are shallow (SEM and AFM). This corresponds to a low-quality scientific report that is not enough to be published in Int. J. Mol. Sci
Answer:
The objective of this study was to synthesize and characterize folic acid-conjugated human serum albumin nanoparticles as targeted carriers for rutin. In a previous study, we investigated and optimized a similar drug delivery system. The novelty of the current work lies in the incorporation of folic acid into the nanoparticle system to enhance cellular uptake. We agree with the fact that a comprehensive physicochemical characterization can offer deeper insight into the nanoparticle system. However, it should also be noted that one of the main conditions for maintaining nanoparticle stability is the use of low concentrations, in order to prevent fibrillization. This, in turn, represents a limitation for characterization techniques such as FTIR and even XPS, which typically require higher sample densities or surface coverage. Concerning the CD or Z potential analysis, we acknowledge theirs importance and hope that funding for such equipments will become available at our institutions in the near future.
We believe there is no doubt that the cell viability results and fluorescence microscopy provide solid experimental evidence supporting albumin-based targeted nanomedicine delivery systems, with clear scientific significance and application potential.
Reviewer 4 Report (New Reviewer)
Comments and Suggestions for Authors
This research provides solid experimental evidence for albumin-based targeted nanomedicine delivery systems and has clear scientific significance and application potential.The author has made comprehensive and detailed revisions in response to the initial review, supplemented key experimental data, and improved language expression, chart quality, and statistical analysis. The current version is rich in data, has clear conclusions, and a rigorous experimental design, fully meeting the publication standards of this journal.
Author Response
Comments and Suggestions for Authors
This research provides solid experimental evidence for albumin-based targeted nanomedicine delivery systems and has clear scientific significance and application potential.The author has made comprehensive and detailed revisions in response to the initial review, supplemented key experimental data, and improved language expression, chart quality, and statistical analysis. The current version is rich in data, has clear conclusions, and a rigorous experimental design, fully meeting the publication standards of this journal.
Thank jou for your evaluation.
Reviewer 5 Report (New Reviewer)
Comments and Suggestions for Authors
The manuscript can be published in present form because "the mentioned additional data were not available" and "the application of other characterization techniques was not possible due to equipment availability and specific experimental conditions".
Author Response
Comments and Suggestions for Authors
The manuscript can be published in present form because "the mentioned additional data were not available" and "the application of other characterization techniques was not possible due to equipment availability and specific experimental conditions".
Thank you for your evaluation and understanding.
Round 3
Reviewer 3 Report (New Reviewer)
Comments and Suggestions for Authors
In its current state the manuscript "Human serum albumin-based nanoparticles for targeted intracellular drug delivery" presents the synthesis of HSA-FA-Ru NPs; the particles were synthesized but not fully characterized, the fact that there was no size characterization of the NPs, besides "to be smaller than 100nm", makes the manuscript not suitable for publication.
Author Response
After carefully revising the version 2 of the manuscript “Human serum albumin-based nanoparticles for targeted intracellular drug delivery” I cannot recommend it publication in Int. J. Mol. Sci. mainly because the characterization of the nanomaterials is incomplete.
C1 Abot the size of the nanoparticles: I my past revision I asked about the discrepancies in the size of the nanoparticles reported by the two techniques, SEM and AFM; the authors responded in lines 144-147 that “Although the values obtained by SEM and AFM are not directly comparable, the two techniques provide complementary information regarding particle morphology and structure. Specifically, AFM measures particle height, while SEM determines the projected diameter.” This mean that they expect that the NPs are not spherical? Because the height and the projected diameter will be in the same ranges for spherical NPS. Then in lines 136-139 “For the HSA NPs, the SEM images obtained (Figure 2A) showed a uniform distribution, these NPs having small sizes of about 50 - 70 nm. Similarly, HSA-FA-Ru NPs (Figure 2B) showed sizes in the tens of nanometers range, while forming aggregates in the hundreds of micrometers range.” But no distribution is shown for these measurements. In lines 127-130 the authors mentioned that “According to literature, HSA NPs dimensions are highly dependent on the processing conditions, such as pH, concentration, temperature or desolvation agent [27]. It is expected that small-sized nano-particles enhance tissue penetration, making HSA and HSA-FA-Ru NPs suitable for targeted delivery of Ru into cancer cells [28, 29].” If the authors recognize the importance of the size for the NPs to act as Ru carriers, how can be acceptable to report a nanomaterial whose size is unknown? I hold my criticism in this part of the manuscript as the size was not measured correctly.
C2 in the past revision I suggested that a more complete characterization must be performed on the HAS and HAS-FA-Ru nanoparticles with techniques such as FT-IR, XPS, UV-Vis, DLS and Z potential will give a better and more complete insight of the NPs physical and chemical properties. No more characterizations were performed to the NPs, and the ones performed are shallow (SEM and AFM). This corresponds to a low-quality scientific report that is not enough to be published in Int. J. Mol. Sci.
Answer:
Considering all the requests and taking into account the logistic limitations regarding the availability of new reagents and experiments involving techniques such as DLS, we provide here an alternative analysis of the nanoparticles size evolution. Specifically, nanoparticle solutions prepared at 200 mg·mL⁻¹ and incubated at 4 °C for 30 days were evaluated by laser diffraction using the FRITSCH ANALYSETTE 22 NanoTec instrument. The results were consistent with those presented in the Supporting Information (SI), S1, and Fig. 3, confirming that at prolonged incubation times or at high concentrations, the nanoparticles tend to form large aggregates, SI Fig. 2.
The results were consistent with those presented in the Supporting Information, Fig. S1, and Fig. 3, confirming that at prolonged incubation times or at high concentrations, the nanoparticles tend to form large aggregates. with Ru forms smaller aggregates around 10 m compared with the sample without Ru, where the magnitude rich more than 500 m) (S1 Fig. 3).
This manuscript is a resubmission of an earlier submission. The following is a list of the peer review reports and author responses from that submission.
Round 1
Reviewer 1 Report
Comments and Suggestions for Authors
- Why are there very few HSA-FA-Ru particles in Figure 1B, AFM images of HSA and HSA-FA-Ru? Authors should have taken similar populated particle images. No Dynamic light scattering data provided.
- In SEM images (Figure 2) for the two types of NPs, there are hardly 3-4 particles visible in HSA nanoparticles. Authors should provide high resolution images with comparable particle populations.
- Figure 3: Which cells used for this experiment? What is the cell density?
- If nanoparticles are toxic to cells below 2 mg/mL concentration, how would authors use 200 mg/mL concentration for Fluorescence microscopy studies? Authors mentioned that “Labeling was performed by incubating NPs of concentration 200 mg/mL with 1 mM thioflavin T in a working volume of 1 mL”. Please justify this statement.
- No IC50 values provided in the manuscript.
- Figure 4: No statistical analyses were performed. Authors should have used GraphPad software to analyze the results. Where are the other controls, like drug alone, FA alone? There are no asterisks (*) observed to show statistical significance in the manuscript.
- What is pink color in Figure 6? What is the evidence for cellular uptake of HSA-FA-Ru loaded nanoparticles in HT-29 cells?
- No details on % drug loading, %encapsulation mentioned in the manuscript. On what basis do authors use FA (0.36 mM), EDC (0.554 mM) and NHS (0.554 mM) to prepare HSA-FA-Ru NPs?
- How do authors know the successful conjunction of Folic acid? What is the evidence to prove it? No NMR, IR, FT-IR, MALDI-TOF data presented.
- Nowhere is mentioned about the nanoparticle release profile.
- Authors have checked the nanoparticles using folate receptor antibodies and western blots to get evidence of folate conjugation on HSA NPs.
Please check the methods. Example, Preparation of HSA NPs. In a first step, HSA-NPs were synthesized by the desolvation method previously described for the preparation of bovine serum albumin nanoparticles [8]. HSA NPs were obtained at room temperature, using EtOH as desolvation agent and glucose as cross-linking agent. Briefly, the pH of HSA (200 mg/mL) solution was established at 8.15, then the sample was stirred (CRS 15X CAPP, Odense, Denmark) at 550 rpm for 10 min. After that, 8 mL EtOH were dropwise.
Author Response
- Why are there very few HSA-FA-Ru particles in Figure 1B, AFM images of HSA and HSA-FA-Ru? Authors should have taken similar populated particle images. No Dynamic light scattering data provided.
Answer:
We appreciate the reviewer’s observation.
Both AFM and SEM techniques require a relatively low surface coverage of nanoparticles in order to properly visualize the substrate and obtain high-resolution images. For AFM in particular, an optimal particle dispersion is essential for accurate morphological analysis. A high density or aggregation of particles may lead to overlapping features, making it difficult to distinguish and accurately measure individual nanoparticles. Therefore, the image in Figure 1B was selected to ensure well-dispersed HSA-FA-Ru particles, suitable for reliable topographical assessment.
Concerning the DLS analysis, we hope that funding for acquisition of such equipment to be available in the near future in our institutions.
- In SEM images (Figure 2) for the two types of NPs, there are hardly 3-4 particles visible in HSA nanoparticles. Authors should provide high resolution images with comparable particle populations.
Answer:
See the answer provided to comment 1.
- Figure 3: Which cells used for this experiment? What is the cell density?
Answer:
As described in the manuscript, no cells were used for this experiment. Fig. 3 represents “Fluorescence microscopy images obtained only for nanoparticles: (A) HSA NPs and (B) HSA-FA-Ru NPs labeled with thioflavin T (ThT) after (A1) and (B1) at 0 h and (A2) and (B2) 24 h incubation, respectively, in aqueous solution”. The meaning of these experiments is now better explained in the revised manuscript:
“Albumin possesses a tertiary structure characterized by a high content of β-sheet mo-tifs, which are often retained during nanoparticle formation. Thioflavin T (ThT) is a benzothiazole dye that exhibits minimal fluorescence in solution but undergoes a dra-matic increase in fluorescence intensity upon binding to β-sheet-enriched structures, particularly those forming cross-β motifs commonly found in amyloid fibrils and simi-lar aggregates. [28]. The mechanism of ThT binding involves its insertion into the grooves formed by the repetitive β-sheet architecture, where restricted intramolecular rotation leads to enhanced quantum yield and fluorescence emission (excitation: ~440–450 nm; emission: ~480–490 nm). This fluorescence enhancement is widely used as an indicator of the presence and spatial distribution of supramolecular β-sheet structures. In this context, ThT labeling serves as a sensitive tool for confirming the structural or-ganization of albumin nanoparticles and for their selective detection under fluores-cence microscopy. Fluorescent labeling of albumin nanoparticles with Thioflavin T enables visualization of β-sheet-rich domains by fluorescence microscopy.Thus, HSA-NPs and HSA-FA-Ru NPs were labeled with the fluorescent indicator thioflavin T (ThT), and abeling was performed by incubating NPs of concentration 200 mg/mL with 1 mM thioflavin T in a working volume of 1 mL, followed by centrifugation and washing.”
- If nanoparticles are toxic to cells below 2 mg/mL concentration, how would authors use 200 mg/mL concentration for Fluorescence microscopy studies? Authors mentioned that “Labeling was performed by incubating NPs of concentration 200 mg/mL with 1 mM thioflavin T in a working volume of 1 mL”. Please justify this statement.
Answer:
See the answer provided to comment 3, in which we mention that this first study focused on nanoparticles without involving cells. Also, a centrifugation step, in which the Thioflavin T was removed is mentioned in the preocedure.
- No IC50 values provided in the manuscript.
Answer:
We appreciate the reviewer’s comment regarding the lack of IC50 values. The primary objective of the present work was the development, synthesis, and physicochemical characterization of folic acid-conjugated human serum albumin nanoparticles (HSA-FA-Ru NPs) as targeted carriers for rutin (Ru). The biological evaluation was focused on demonstrating the selective cytocompatibility and preliminary therapeutic potential of these nanocarriers. Determination of IC50 values requires detailed dose-response studies with a broader range of concentrations and replicates, which go beyond the exploratory scope of this study. The optimization of treatment concentrations aimed at selectively reducing the viability of cancer cells while preserving the integrity of healthy cells is currently being planned within the framework of future research projects, pending funding support. We have now added a clarifying statement in the revised manuscript to acknowledge this limitation and outline our future directions, but we are aware of the importance of determining the IC₅₀ values. Such determinations require a broader range of concentrations, higher resolution dose-response curves, and advanced statistical modeling to accurately assess half-maximal inhibitory concentrations.
Following your question and observation, the text has been completed as follows:
’’The focus of this work was the successful development and characterization of a folic acid-functionalized HSA-based delivery system and its preliminary evaluation in vitro. Future studies will address the optimization of nanoparticle formulations and treatment conditions, aiming to define therapeutic concentration windows that ensure maximum cytotoxicity toward cancer cells while preserving normal cell viability. Our current results provide strong evidence of the differential cytotoxic effect of HSA-FA-Ru nanoparticles—marked by a significant reduction in HT-29 cell viability and minimal impact on L929 fibroblasts.’’
- Figure 4: No statistical analyses were performed. Authors should have used GraphPad software to analyze the results. Where are the other controls, like drug alone, FA alone? There are no asterisks (*) observed to show statistical significance in the manuscript.
Answer:
The statistical significance was introduced for Fig. 4 of the manuscript. For statistical analyses, ANOVA with Tukey post-hoc tests were performed.
- What is pink color in Figure 6? What is the evidence for cellular uptake of HSA-FA-Ru loaded nanoparticles in HT-29 cells?
Answer:
Thank you for your questions. Indeed, the legend of Figure 6 do not explain the nature of pink color and in the revised manuscript this aspect was corrected and the manuscript was modified as follows:
“For the experiments on HT-29 cells actin filaments were labeled with a red-emission fluorophore conjugated with phalloidin. At the same time, the rich beta sheet secondary structure of albumin nanoparticles was labeled with thioflavin T (green emission). Therefore, combining both red and green results in pink.”
- No details on % drug loading, %encapsulation mentioned in the manuscript. On what basis do authors use FA (0.36 mM), EDC (0.554 mM) and NHS (0.554 mM) to prepare HSA-FA-Ru NPs?
Answer:
These experimental conditions are the result of previous experiments to optimize the method of obtaining nanoparticles, with a balance between characteristics such as size, stability on the one hand, and the efficiency of folic acid (FA) conjugation on the surface of human serum albumin (HSA) and the maintenance of the structural integrity of the nanoparticles on the other hand, crucial aspects for the subsequent loading and release of rutin (Ru).
- How do authors know the successful conjunction of Folic acid? What is the evidence to prove it? No NMR, IR, FT-IR, MALDI-TOF data presented.
Answer:
Thank you for your observation. Although ideally we would have liked to include spectral data, we were able to assess (even indirectly) the conjugation, based on the following two aspects: 1) the use of a well-established FA-protein conjugation protocol from the literature, which has already been well characterized and validated in previous studies (Chilom et al., Coatings, 2023), work also cited in this article, and 2) an indirect evidence comes from in vitro experiments,which show a significantly higher cellular internalization of HSA-FA-Ru nanoparticles into target cells compared to unmodified HSA nanoparticles This is a strong evidence for the presence of functional FA on the nanoparticles surface.
- Nowhere is mentioned about the nanoparticle release profile.
Answer:
The focus of this work was the successful development and characterization of a folic acid-functionalized HSA-based delivery system and its preliminary evaluation in vitro. Although the nanoparticle release profile represents an important event, this wasn’t the aim of our work. Probably, once internalized via folic acid receptor-mediated or fluid-phase endocytosis, albumin nanoparticles will be subjects of enzymatic cleavage and the partial unfolding and degradation of the protein matrix will finally lead to the release of encapsulated rutin.
- Authors have checked the nanoparticles using folate receptor antibodies and western blots to get evidence of folate conjugation on HSA NPs?
Answer:
This is a very good observation. At the moment, it was not possible to use folate receptor antibodies nor the Western Blot technique as methods to confirm the conjugation of folic acid (FA) to human serum albumin (HSA) nanoparticles. As this study will be continued in the future, we will try to include these approaches for future validation.
Comments on the Quality of English Language. Please check the methods. Example, Preparation of HSA NPs. In a first step, HSA-NPs were synthesized by the desolvation method previously described for the preparation of bovine serum albumin nanoparticles [8]. HSA NPs were obtained at room temperature, using EtOH as desolvation agent and glucose as cross-linking agent. Briefly, the pH of HSA (200 mg/mL) solution was established at 8.15, then the sample was stirred (CRS 15X CAPP, Odense, Denmark) at 550 rpm for 10 min. After that, 8 mL EtOH were dropwise.
Answer:
The manuscript has been revised in terms of English writing.

Reviewer 2 Report
Comments and Suggestions for Authors
Dear Authors,
The manuscript ijms-3617098 titled “Human serum albumin-based nanoparticles for targeted intracellular drug delivery” delineates study findings concerning the synthesis and characterization of folic acid conjugated human serum albumin nanoparticles loaded with rutin for targeted delivery into two cell lines, L929 and HT-29.
The title is informative and concise. The manuscript is well-structured and comprises the following sections: Abstract, Introduction, Results and Discussion, Materials and Methods, and Conclusions.
The abstract summarizes the main points of the paper.
The introduction provides general information for colorectal cancer, and attempts of researchers to create nanoparticles based on serum proteins, as well as the anticancer properties of flavonoids. The authors have motivated very well research purposes.
The methods used are appropriate for the study and detailed description is provided.
The results are described in details and clearly presented in the main text and in the supplementary files. They are thoroughly analyzed and discussed in the main text.
The conclusions are rational and based on the article's content. The referenced literature is appropriate and relevant to this research.
I have the following remarks addressed to you.
1) Line 95 - “…ionic straight..” – it seems to be a typing error
2) Lines 165,166 – “..lower viability was obtained for the L929 line, about 70% (Figure 5A (●)), for NPs concentrations of 1 mg/mL and very low, around 50%, for the HT-29 cell line (Figure 4B (●)).” - For HT29 cell line viability is 50% at 2 mg/mL concentration as it is seen on the figure 4B – Did you mean 2mg/mL concentration?
3) Line 302 – MTT or MTS assay as it is sat in line 157, because they work a little differently
4) Line 336 – “Figure 4” – Isn’t it Figure 1A?
5) Line 337 “…after about 20 days, the stability of HSA NPs is 337 54.9% and that of HSA-FA-Ru NPs is 68.8%.” – From the figure it seems to be the other way around.
Overall, the study is well designed and conducted and the findings demonstrated that human serum albumin nanoparticles are biocompatible and could serve as a platform for targeted drug delivery.
Author Response
Comments and Suggestions for Authors
Answer:
Thank you for your appreciation and constructive comments.
- Line 95 - “…ionic straight..” – it seems to be a typing error
Answer:
We have corrected in ionic strength
- Lines 165,166 – “..lower viability was obtained for the L929 line, about 70% (Figure 5A (●)), for NPs concentrations of 1 mg/mL and very low, around 50%, for the HT-29 cell line (Figure 4B (●)).” - For HT29 cell line viability is 50% at 2 mg/mL concentration as it is seen on the figure 4B – Did you mean 2mg/mL concentration?
Answer:
For a better understanding, the manuscript was modified as follows.
“By using HSA-FA-Ru NPs, a slightly decrease of viability, but still elevated, was obtained for the L929 line, about 70% (Figure 4A (●)), for NPs concentrations above 1 mg/mL, whereas higher concentrations (around 2 mg/mL) of NPs induced a decrease of viability, below 50% (Figure 4B (●)), for the HT-29 cell line.”
- Line 302 – MTT or MTS assay as it is sat in line 157, because they work a little differently
Answer:
Indeed, it is the MTS test. Thank you for your observation. We have corrected in MTS
- Line 336 – “Figure 4” – Isn’t it Figure 1A?
Answer:
Thank you for your observation. We have corrected in Figure S1.
Now, the text looks like this:
“results obtained are shown in Figure S1“
- Line 337 “…after about 20 days, the stability of HSA NPs is 337 54.9% and that of HSA-FA-Ru NPs is 68.8%.” – From the figure it seems to be the other way around.
Answer:
Thank you for your observation. Indeed, the values have been reversed. The manuscript was modified according with the correct values:
“after approximately 20 days, the stability of HSA nanoparticles was 68.8%, while that of HSA-FA-Ru nanoparticles was 54.9%.“.
